# Mitigating Occlusions in Virtual Try-On via A Simple-Yet-Effective Mask-Free Framework

**Chenghu Du**[1]   **Shengwu Xiong**[2]   **Junyin Wang**[1]   **Yi Rong**[1*]   **Shili Xiong**[1,3*]

[1]School of Computer Science and Artificial Intelligence, Wuhan University of Technology
[2]Interdisciplinary Artificial Intelligence Research Institute, Wuhan College
[3]Shanghai Artificial Intelligence Laboratory

{duch, xiongsw, wjy199708, yrong}@whut.edu.cn   slxiong.illinois@gmail.com
https://du-chenghu.github.io/OccFree-VTON/

## Abstract

This paper investigates the occlusion problems in virtual try-on (VTON) tasks. According to how they affect the try-on results, the occlusion issues of existing VTON methods can be grouped into two categories: (1) Inherent Occlusions, which are the ghosts of the clothing from reference input images that exist in the try-on results. (2) Acquired Occlusions, where the spatial structures of the generated human body parts are disrupted and appear unreasonable. To this end, we analyze the causes of these two types of occlusions, and propose a novel mask-free VTON framework based on our analysis to deal with these occlusions effectively. In this framework, we develop two simple-yet-powerful operations: (1) The background pre-replacement operation prevents the model from confusing the target clothing information with the human body or image background, thereby mitigating inherent occlusions. (2) The covering-and-eliminating operation enhances the model's ability of understanding and modeling human semantic structures, leading to more realistic human body generation and thus reducing acquired occlusions. Moreover, our method is highly generalizable, which can be applied in in-the-wild scenarios, and our proposed operations can also be easily integrated into different generative network architectures (e.g., GANs and diffusion models) in a plug-and-play manner. Extensive experiments on three VTON datasets validate the effectiveness and generalization ability of our method. Both qualitative and quantitative results demonstrate that our method outperforms recently proposed VTON benchmarks.

## 1 Introduction

Virtual Try-On (VTON) technology [1, 2, 3, 4, 5, 6, 7, 8, 9] aims to synthesize the user's desired clothing from a garment image onto human models or real-person images. It provides customers with a convenient and efficient try-on experience that greatly reduces the time and effort required in traditional offline shopping modes. Therefore, VTON has gained substantial interest in recent years, particularly in the fields of fashion, e-commerce, and entertainment.

One of the main problems in building a VTON model lies in the unpaired nature of the training data. While the target clothing image and the corresponding ground-truth image of a person wearing this clothing are provided, the input reference image that contains the same person but wearing a different clothing is typically unavailable. According to the strategies adopted to address this issue, current VTON methods can be roughly divided into two categories: (1) Mask-based methods [10, 11, 5, 12, 7, 8, 13] consider VTON as a self-supervised image inpainting problem. They first

---

[*]Corresponding authors.

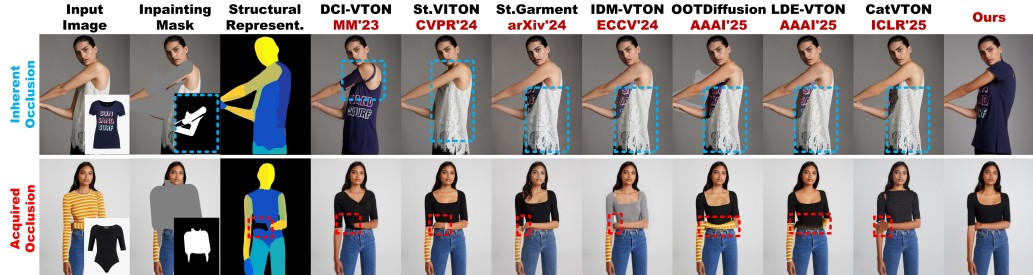

Figure 1: **Visualization of the occlusion issues in VTON and the effectiveness of the proposed method.** It illustrates the inherent occlusion (Cyan regions) caused by imprecise inpainting masks and acquired occlusion (Red regions) resulting from erroneous human structural representations.

mask the clothing-relevant regions in the ground-truth image, and then attempt to recover these regions based on the target clothing image. In contrast, (2) Mask-free methods [1, 2, 3, 14, 15, 16] directly synthesize pseudo reference images (*e.g.*, via generative model) as the input for VTON model training, thus eliminating the requirement of generating inpainting masks in inference phase.

However, during the training stage, these two types of methods both rely on the inpainting masks to accurately remove the clothing regions or generate appropriate pseudo reference inputs, respectively. *As a result, the imprecise masks that cannot fully cover the clothing regions will incorrectly remain a portion of the target clothing areas in the masked (or pseudo) input images.* Such remained regions can be misidentified as the human body or background and be mistakenly preserved. Therefore, training with these samples will establish incorrect associations between the target clothing information and the human body or background pixels. This will finally lead to a sub-optimal model that outputs the try-on results containing ghosts of the clothing from reference input images, which are recognized as **Inherent Occlusions** [17] (see the top row in Fig. 1). In addition, existing VTON approaches may also suffer from another type of **Acquired Occlusions**, *which is typically caused by erroneous human structural representations (HSRs) that misguide the learning and inference processes of the generator.* Consequently, the spatial structures of human body parts in the try-on results will be disrupted and appear unreasonable, as shown in the bottom row of Fig. 1.

In this study, we propose a simple-yet-effective approach that is able to deal with both inherent and acquired occlusions in a unified mask-free VTON framework. On the one hand, to tackle inherent occlusions, we design a **background pre-replacement** operation that replaces the image regions outside the combination of the inpainting mask and the person identity mask in each training sample with either a pure background or a random scene image. In this way, the remained target clothing areas will be filled with background pixels, thus allowing the learned model to differentiate the clothing information from the background more effectively. On the other hand, since HSRs mainly affect the generative quality of human body parts, the acquired occlusions caused by erroneous HSRs mostly occur in situations where the human body generation is required, i.e., when trying on clothes with smaller body coverage to the reference image with a larger clothing area. To this end, we develop a **covering-and-eliminating** operation to mimic these situations so that the related robustness can be improved. Specifically, it first produces pseudo reference images with a different clothing that completely covers the original target clothing. Then, our model is trained to eliminate these coverings and reconstruct the underlying human body areas. During this process, the model's understanding of human semantic structures will be enhanced under the accurate supervision of real ground-truths, thereby mitigating the negative effects of incorrect HSRs. Extensive experiments on three VTON datasets validate the effectiveness of our framework, surpassing recently proposed benchmarks both qualitatively and quantitatively. The main contributions of this paper are summarized as follows:

- We analyze the causes of both inherent and acquired occlusions, and propose a novel mask-free VTON framework that can effectively handle these two types of occlusions.

- We design a background pre-replacement operation to prevent the model from mistaking the target clothing information as the human body or image background during the training process, thus mitigating inherent occlusions.

- We design a covering-and-eliminating operation to generate pseudo reference images that mimic the situations where acquired occlusions mostly occur. Under the supervision of real

ground-truths, eliminating the larger clothing in these reference images and reconstructing the human body will improve the model's ability of modeling human semantic structures.

- We validate the effectiveness of our method through extensive experiments and demonstrate its scalability to be compatible with different generative network architectures.

## 2 Related Work

**Virtual Try-Ons.** Recent work has achieved impressive results in virtual try-ons. For instance, Xie *et al.* [4] then proposed GP-VTON, which warps garments based on the characteristics of each area of garments, advancing garment alignment to the current peak of fully generalizable performance. However, it heavily relies on human parsing, and mask errors lead to a series of failed results. To address this, Du *et al.* [15] designed a cyclic architecture USC-PFN that successfully eliminates the negative impact of masks during inference. Recently, methods based on diffusion models have achieved notable results. For instance, DCI-VTON [5], based on the PbE architecture [11], pioneered the possibility of achieving high-quality virtual try-ons. Subsequently, methods like LaDI-VTON [10], StableVITON [7], and AnyDoor [13] have focused on the diffusion model structure to enhance generation quality. However, it has been demonstrated that even elementary Latent Diffusion Models are capable of producing high-quality results. Therefore, optimizing the final output results seems to be more effective than improving the network structure. Recently, Chong *et al.* [9] proposed CatVTON. They found that training only the self-attention layers of the diffusion model can achieve model convergence at an extremely low computational cost. This undoubtedly provides a more efficient research path for subsequent studies. Since then, it seems that the design of model architecture is a laborious and unprofitable task. Based on this, we propose a framework that focuses on the needs of the task itself rather than the design of model structure. It can easily solve the occlusion problem that all current methods have failed to address, and provides a new shortcut for future research on more robust and practical virtual try-on.

**Occlusion.** This is a long-standing challenge in virtual try-on, as the unpaired nature of training data and the imprecise segmentation masks often lead to two types of occlusions (inherent occlusion and acquired occlusion). Early works used 3D models [18] or dense pose [19] to infer missing regions, while recent methods adopt inpainting networks conditioned on segmentation masks [17]. Despite these advances, most existing methods treat occlusion as a passive artifact to be inpainted, rather than actively modeling the interaction between visible and occluded regions. We propose a simple-yet-effective mask-free framework that eliminates both types of occlusions via two key operations: background pre-replacement and covering-and-eliminating, which enhances the model's understanding of human semantic structures.

## 3 Preliminary

Given a VTON dataset $\{(\mathbf{g}_i, \mathbf{p}_i)\}_i \in \mathbf{D}$, where $\mathbf{g}_i \in \mathbb{R}^{3 \times H \times W}$ represents source clothing images and $\mathbf{p}_i \in \mathbb{R}^{3 \times H \times W}$ denotes reference person images depicting individuals wearing the corresponding clothing from $\mathbf{g}_i$. Due to the unpaired nature of the training data, *i.e.* the input $\mathbf{p}_i$ that contains the same person but wearing a different clothing is typically unavailable, prior methods [4, 5, 7, 8] typically leverage a mask-based inpainting model $\mathcal{M}_\phi$ parameterized by $\phi$, which formulate VTON as an optimization problem by minimizing the following training objective:

$$\phi^* = \arg\min_\phi \ \mathcal{L}_{\text{dist}}\bigg(\mathbf{p}_i, \mathcal{M}_\phi\big(\mathbf{m}_{\text{agn}_i}, \mathbf{p}_{\text{agn}_i}, \mathbf{g}_i\big)\bigg), \tag{1}$$

where $\mathbf{m}_{\text{agn}_i} \in \{0, 1\}^{H \times W}$ is a inpainting mask, which represents the area of the clothing to be changed on $\mathbf{p}_i$ as well as the adjacent skin regions. $\mathbf{p}_{\text{agn}_i} = \mathbf{p}_i \odot (1 - \mathbf{m}_{\text{agn}_i})$ represents an inpainting person. Both are shown in Fig. 1, column 2. $\mathcal{L}_{\text{dist}}(\cdot)$ denotes a suitable loss function, *e.g.*, $\ell_1$ and MSE losses. To compensate for the missing structural information (*e.g.*, arm shape) in $\mathbf{m}_{\text{agn}_i}$ region, additional human structural representations $\mathbf{r}_i$ (*e.g.*, densepose map [19]) of $\mathbf{p}_i$ are typically introduced as supplementary conditions to guide the synthesis of limbs and clothing. However, the low quality of $\mathbf{m}_{\text{agn}_i}$ and $\mathbf{r}_i$ usually hinders inpainting methods from obtaining the ideal semantic information to accurately translate the desired human body regions [1, 2], resulting in defective appearance **occlusion**, as depicted in Fig. 1.

Table 1: **Comparison of occlusion handling capabilities** between mask-based and mask-free VTONs. $\mathcal{M}_\phi$ and $\mathcal{M}_\psi$ correspond to mask-based and mask-free models. $\mathbf{D}_{\mathrm{syn}}$ and $\mathbf{D}_{\mathrm{real}}$ denote synthetic (by pre-trained $\mathcal{M}_\phi$) and real datasets. $\mathbf{D}^*_{\mathrm{syn}}$ is generated from well-trained $\mathcal{M}_\psi$.

| Model | Training Data | Mask-Free | Inherent Occlusion | Acquired Occlusion |
|-------|---------------|-----------|--------------------|--------------------|
| $\mathcal{M}_\phi$ | $\mathbf{D}_{\mathrm{real}}$ | ✗ | ✗ | ✗ |
| $\mathcal{M}_\psi$ | $\mathbf{D}_{\mathrm{syn}}, \mathbf{D}_{\mathrm{real}}$ | ✓ | ✗ | ✗ |
| $\mathcal{M}_\theta$ | $\mathbf{D}^*_{\mathrm{syn}}, \mathbf{D}_{\mathrm{real}}$ | ✓ | ✓ | *partial* |

To mitigate the negative impacts of $\mathbf{m}_{\mathrm{agn}_i}$ and $\mathbf{r}_i$, a mask-free architecture is proposed [1, 2, 3]. It uses the pre-trained well-performed, mask-based inpainting model $\mathcal{M}^*_\phi$ to generate a pseudo reference image $\mathbf{t}_{\mathrm{un}_i} \in \mathbb{R}^{3 \times H \times W}$ ($\mathbf{p}_i$ wearing random clothing $\mathbf{g}_{\mathrm{un}_i}$), thus replacing the functions of $\mathbf{m}_{\mathrm{agn}_i}$ and $\mathbf{r}_i$, to form data triplets $(\mathbf{t}_{\mathrm{un}_i}, \mathbf{g}_i, \mathbf{p}_i)$ for the full-supervised training of a mask-free $\mathcal{M}_\psi$ parameterized by $\psi$. It can also be formulated as minimizing the following training objective:

$$\psi^* = \arg\min_\psi \; \mathcal{L}_{\mathrm{dist}}\left( \mathbf{p}_i, \mathcal{M}_\psi \underbrace{\left( \mathcal{M}^*_\phi\left(\mathbf{m}_{\mathrm{agn}_i}, \mathbf{p}_{\mathrm{agn}_i}, \mathbf{g}_{\mathrm{un}_i}, \mathbf{r}_i\right)}_{\texttt{Pseudo Reference Image } \mathbf{t}_{\mathrm{un}_i}}, \mathbf{g}_i\right) \right). \tag{2}$$

Since $\mathbf{m}_{\mathrm{agn}_i}$ and $\mathbf{r}_i$ are no longer required as input conditions at $\mathcal{M}_\psi$, it seems that this architecture can thoroughly block the negative impact of $\mathbf{m}_{\mathrm{agn}_i}$ and $\mathbf{r}_i$ on the results. However, as can be seen from Eq. (2), the pseudo reference image $\mathbf{t}_{\mathrm{un}_i}$ related to defective $\mathbf{m}_{\mathrm{agn}_i}$ and $\mathbf{r}_i$ that is not responsible in $\mathcal{M}^*_\phi$ can still be transferred to $\mathcal{M}_\psi$ during training, which can still cause **occlusion** in the results.

## 4  Occlusion Analysis

As can be seen from Fig. 1, occlusion manifests in different forms, but it is mainly caused by $\mathbf{m}_{\mathrm{agn}}$ and $\mathbf{r}$ (subscript $i$ is omitted for brevity). We refer to the occlusion caused by $\mathbf{m}_{\mathrm{agn}}$ as "**inherent occlusion**," while the occlusion caused by $\mathbf{r}$ is called "**acquired occlusion**."

● **Inherent Occlusion.** The inherent occlusion stems from the *erroneous parsing of the clothing-relevant region* in $\mathbf{p}$, leading to an imprecise inpainting mask $\mathbf{m}_{\mathrm{agn}}$ that missegments the human body, background, and clothing regions of $\mathbf{p}$. For mask-based methods, $\mathbf{m}_{\mathrm{agn}}$ serves as the guiding condition for the inpainting model $\mathcal{M}_\phi$ (in Eq. (1)) to determine the area $\mathbf{p}_{\mathrm{agn}} = \mathbf{p} \odot (1 - \mathbf{m}_{\mathrm{agn}})$ that is to be preserved before and after trying on. Therefore, the residual source clothing areas in $\mathbf{p}_{\mathrm{agn}}$ were treated as immutable and preserved unconditionally by $\mathcal{M}_\phi$. While ensuring accurate inpainting masks for both training data and inference samples might seem like a solution, the diversity of people, clothing, and scenes worldwide makes it impossible to guarantee consistently accurate masks during large-scale data training and inference. Therefore, this also directly sows the seeds of trouble for mask-free methods. In mask-free methods, the mask-based inpainting model $\mathcal{M}_\phi$ must provide a pseudo-reference (person) image $\mathbf{t}_{\mathrm{un}}$ for model $\mathcal{M}_\psi$ (in Eq. (2)) to form a data triplet $\{(\mathbf{t}_{\mathrm{un}}, \mathbf{g}), \mathbf{p}\}$, enabling fully supervised training for $\mathcal{M}_\psi$. This reconstructs $\mathbf{p}$ without needing the inpainting mask $\mathbf{m}_{\mathrm{agn}}$. However, $\mathbf{t}_{\mathrm{un}}$ contains residual areas from $\mathbf{p}$, which are mistakenly treated as background during reconstruction and mapped unchanged into the results. This creates incorrect associations between the target clothing and the human body or background pixels, ultimately leading to sub-optimal models that output try-on results with ghosts of the clothing from reference images. Consequently, during inference, when dealing with complex postures or rare clothing styles in person image $\mathbf{p}$, the mask-free $\mathcal{M}_\psi$ still misrecognizes boundaries between the human body and background, resulting in inherent occlusion.

● **Acquired Occlusion.** The acquired occlusion stems from *erroneous parsing of the human structure*, resulting in an inaccurate human structural representations (HSRs) $\mathbf{r}$ that misguide the learning and inference processes of the generator. For example, if a human parsing (semantic segmentation) map with a missing arm is used to translate into a human image, it is evident that the arm in the resulting image will also be missing. Therefore, as one of the conditional inputs, the quality of HSRs determines the quality of the generated human regions. In mask-based and mask-free methods, the manner in which acquired occlusion is produced is the same as that of inherent occlusion. The only difference is that acquired occlusion directly affects the model's understanding of human semantic

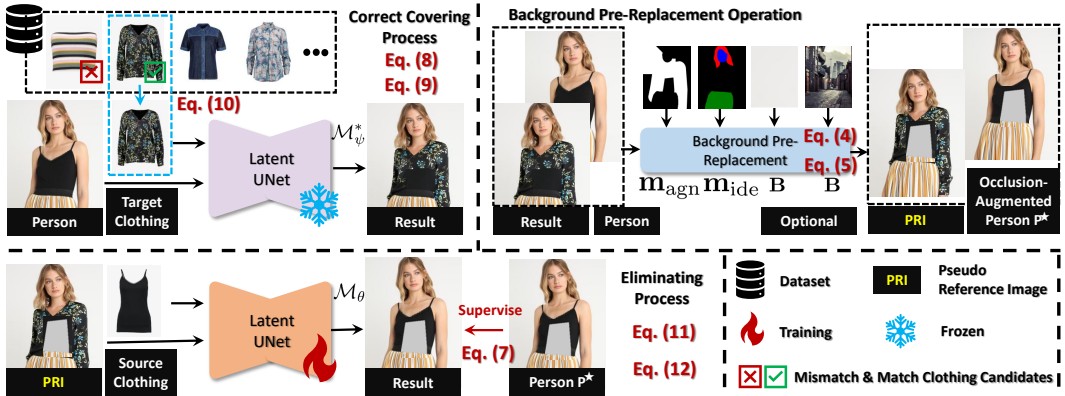

Figure 2: **Overview of our proposed framework.** It includes two operations: the covering and eliminating processes to integrate clothing with a person image, and the background pre-replacement operation to create a pseudo reference image by substituting the original background.

structures, thereby translating into disruptions and rendering the spatial structure of clothing and body parts unreasonable.

Based on the above analysis, we summarize the occlusion handling capabilities of mask-based and mask-free VTON methods in Tab. 1. It can be seen that they are all directly or indirectly affected by $\mathbf{m}_{\mathrm{agn}}$ and $\mathbf{r}$. Motivated by the desire to break through the limitations faced by the aforementioned VTON methods, our goal is to design a new framework with a robust VTON model $\mathcal{M}_\theta$ to fundamentally mitigate the two types of occlusion issues.

# 5 Proposed Framework

To address the two types of occlusion problems, we propose a novel VTON framework (see Fig. 2) that aims to achieve high-quality occlusion-free VTON through a simple-yet-effective method.

## 5.1 How to Eliminate Inherent Occlusion ?

To eliminate inherent occlusion, a potential and feasible solution is to simultaneously *remove the residual regions from both the inputs and the supervision signals during training.* This prevents the model from mistakenly identifying these residual regions as part of the background or human body. However, detecting and segmenting the residual regions still involves the unreliability of segmentation errors. Therefore, we attempt **to replace the entire background that contains the residual regions.**

**Background Pre-Replacement.** To this end, we propose a background pre-replacement operation to intercept the residual regions that may propagate during the training process from the source. Given a data group $(\mathbf{p}, \mathbf{m}_{\mathrm{agn}}, \mathbf{m}_{\mathrm{ide}}) \in \mathbf{D}_{\mathrm{real}}$, where $\mathbf{m}_{\mathrm{ide}} \in \{0,1\}^{H \times W}$[2] is a person identity mask from human parsing, *e.g.*, the head, hands, and feet. As shown in Figs. 2, we segment out the residual background mask $\mathbf{m}_{\mathrm{RB}} \in \{0,1\}^{H \times W}$ from $\mathbf{m}_{\mathrm{agn}}$ via Eq. (3):

$$\mathbf{m}_{\mathrm{RB}} = 1 - \big(\mathbf{m}_{\mathrm{agn}} + \mathbf{m}_{\mathrm{ide}} \odot (1 - \mathbf{m}_{\mathrm{agn}})\big), \tag{3}$$

where $\odot$ represents the element-wise (Hadamard) product. In this case, $(1 - \mathbf{m}_{\mathrm{RB}}) \odot \mathbf{p}$ contains not only the *background* but also *residual clothing regions*. Then, we use a background map $\mathbf{B} \in \mathbb{R}^{3 \times H \times W}$ to replace $(1 - \mathbf{m}_{\mathrm{RB}}) \odot \mathbf{p}$ with a completely pure background, thereby completely severing the connection between the background and the human body region, expressed in Eq. (4):

$$\mathbf{p}^\star = \mathbf{m}_{\mathrm{RB}} \odot \mathbf{p} + (1 - \mathbf{m}_{\mathrm{RB}}) \odot \mathbf{B}, \tag{4}$$

where $\mathbf{p}^\star$ is the background-replaced version of $\mathbf{p}$. Thus, $\mathbf{p}$ ensures that there are no residuals in the supervision signal. For the input side, mask-based methods, which require the input of $\mathbf{m}_{\mathrm{agn}}$, are

---

[2]The implementation method is shown in the **Technical Appendices** (Sec. C)

unable to effectively eliminate occlusions. Therefore, we use $\mathbf{m}_{\mathrm{RB}}$ to process its result $\mathbf{t}_{\mathrm{un}}$, thereby removing the residuals in the input of the mask-free model, as shown in Eq. (5):

$$\mathbf{t}_{\mathrm{un}}^{\star} = \mathbf{m}_{\mathrm{RB}} \odot \mathbf{t}_{\mathrm{un}} + (1 - \mathbf{m}_{\mathrm{RB}}) \odot \mathbf{B}. \tag{5}$$

For specific improvements to the mask-free methods, please refer to Eq. (22). In summary, our design aims to enable the model to accurately **distinguish** between the human body and background from $\mathbf{p}$.

**In-the-Wild Scenes.** Currently, the datasets in use often feature meaningless and low-diversity scenarios. For instance, the backgrounds in the VITON [20] and VITON-HD [21] datasets are irregularly grayish-white, and most backgrounds in the DressCode dataset [22] are just monotonous wall colors. To enable the model to focus on changes in the inpainting region without being distracted by the diverse backgrounds found in real-world scenarios, the background map $\mathbf{B}$ can be designed to be *random* in-the-wild scene images generated by a pre-trained well-performed T2I (Text-to-Image) model [23, 24]: $\mathbf{B} = \mathtt{T2I}(\mathbf{c}, \boldsymbol{\epsilon})^{3}$, where $\mathbf{c}$ is the prompt and $\boldsymbol{\epsilon}$ is random Gaussian noise. Alternatively, $\mathbf{B}$ can be designed as *random* single value matrix: $\mathbf{B} = \mathtt{random}(0, 1) \in \mathbb{R}^{3 \times H \times W}$ for the pure background scenes.

### 5.2 How to Eliminate Acquired Occlusion ?

To address acquired occlusion as much as possible, one potential solution is *to significantly enhance the model's ability to understand human semantic structures*. To this end, a straightforward idea is **to first produce pseudo reference images with a different clothing that completely covers the original target clothing in $\mathbf{p}$. Then, our model is trained to eliminate these coverings and reconstruct the underlying human body areas.** During this process, the model's understanding of human semantic structures will be enhanced under the accurate supervision of real ground-truths, thereby mitigating the acquired occlusions of incorrect HSRs.

**Covering and Eliminating Processes.** We introduce an ideal architecture that can achieve the desired processes with a de-occlusion model $\mathcal{M}_{\theta}$, parameterized by $\theta$, by minimizing the following training objectives:

$$\theta^{*} = \arg\min_{\theta} \; \mathcal{L}_{\mathrm{dist}}\bigg( \mathbf{p}^{\star}, \mathcal{M}_{\theta}\Big( \underbrace{\mathcal{M}_{\theta}\big(\mathbf{p}^{\star}, \mathbf{g}_{\mathrm{un}}\big)}_{\texttt{Covering Process}}, \mathbf{g} \Big) \bigg). \tag{6}$$

$$\underbrace{\phantom{\theta^{*} = \arg\min_{\theta} \; \mathcal{L}_{\mathrm{dist}}\bigg( \mathbf{p}^{\star}, \mathcal{M}_{\theta}\Big( \mathcal{M}_{\theta}\big(\mathbf{p}^{\star}, \mathbf{g}_{\mathrm{un}}\big), \mathbf{g} \Big) \bigg)}}_{\texttt{Eliminating Process}}$$

It can be observed that the architecture completely discards HSRs and instead acquires human structural information by learning to parse the human body within $\mathbf{p}^{\star}$ itself. However, due to the absence of corresponding ground truth for $(\mathbf{p}^{\star}, \mathbf{g}_{\mathrm{un}})$, this architecture fails to converge. Moreover, since the mask-free model does not have the interference of $\mathbf{m}_{\mathrm{agn}}$ and $\mathbf{r}$ present in the mask-based model, we positively believe that *the cover results of mask-free model $\mathcal{M}_{\psi}^{*}$ can serve as the more beneficial pseudo reference images* $\mathbf{p}_{\mathrm{un}}$. To leverage this, we directly use the results of $\mathcal{M}_{\psi}^{*}$ to optimize $\mathcal{M}_{\theta}$. Therefore, Eq. (6) is rewritten as Eq. (7):

$$\theta^{*} = \arg\min_{\theta} \; \mathcal{L}_{\mathrm{dist}}\big(\mathbf{p}^{\star}, \mathcal{M}_{\theta}\big(\mathbf{p}_{\mathrm{un}}, \mathbf{g}\big)\big), \; \text{where} \; \mathbf{p}_{\mathrm{un}} = \mathtt{BPR}(\mathcal{M}_{\psi}^{*}(\mathbf{p}, \mathbf{g}_{\mathrm{un}})), \tag{7}$$

where $\mathtt{BPR}$ is background pre-replacement operations. However, the given $\mathbf{g}_{\mathrm{un}}$ is random, whereas the $\mathbf{g}_{\mathrm{un}}$ we expect here should have an area larger than that of the clothing $\mathbf{g}$ worn by $\mathbf{p}^{\star}$.

**Correct Covering Process.** For simplicity and readability, we first analyze the **upper** clothing ($\mathbf{g}$, $\mathbf{g}_{\mathrm{un}}$) here, while other cases, such as dresses, are discussed in the **Technical Appendices** (Sec. C.2). To obtain the desired $\mathbf{g}_{\mathrm{un}}$, inspired by [6], we design a functional function $\mathcal{A}(\cdot)$ to determine the size of the clothing $\mathbf{g}$ on $\mathbf{p}^{\star}$ and the size of the clothing $\mathbf{g}_{\mathrm{un}}$. Although directly comparing sizes is intuitive and effective, for upper clothing, there may be cases where the sleeves of $\mathbf{g}_{\mathrm{un}}$ are longer than those of $\mathbf{g}$, but the neckline is lower than that of $\mathbf{g}$. This makes it impossible to accurately judge the size and achieve complete coverage. Therefore, we conduct a more accurate indirect comparison by separately comparing the arm length and neckline size of $\mathbf{p}^{\star}$ and $\mathbf{p}_{\mathrm{un}}$. To this end, we trained a

---

[3]The implementation method is shown in the **Technical Appendices** (Sec. C)

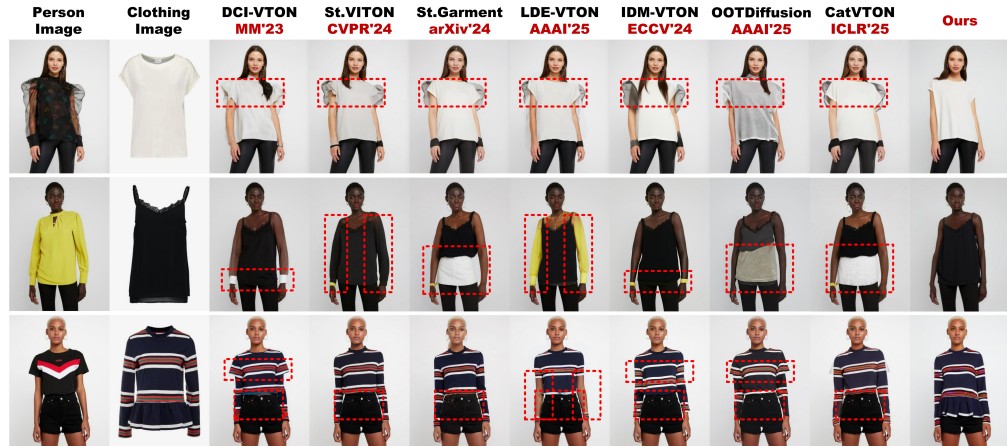

| Person Image | Clothing Image | DCI-VTON MM'23 | St.VITON CVPR'24 | St.Garment arXiv'24 | LDE-VTON AAAI'25 | IDM-VTON ECCV'24 | OOTDiffusion AAAI'25 | CatVTON ICLR'25 | Ours |

Figure 3: **Qualitative results** on the VITON-HD dataset. The baseline methods consist of seven SOTA diffusion-based methods. Red dashed boxes highlight the limitations of each method.

parser $S$, which can generate human parsing (semantic segmentation) maps $\mathbf{m}^\mathrm{t} \in \mathbb{R}^{18 \times H \times W}$ for 18 human semantic categories, thereby segmenting the human body region with specified semantics,

$$\mathbf{m}^\mathrm{t} = S(\mathbf{p}_\mathrm{un}), \text{ where } S^* = \arg\min_S \ \|\mathbf{m}^\mathrm{s} - S(\mathbf{p}^\star)\|_2^2, \tag{8}$$

here, $\mathbf{m}^\mathrm{s} \in \mathbf{D}_\mathrm{real}$ is the human parsing map of $\mathbf{p}^\star$. Therefore, $\mathcal{A}(\cdot)$ is formulated as Eq. (9):

$$\mathcal{A}(\mathbf{m}^\mathrm{s}_j, \mathbf{m}^\mathrm{t}_j) = \sum_{w=1}^{W} \sum_{h=1}^{H} (\mathbf{m}^\mathrm{t}_j - \mathbf{m}^\mathrm{s}_j \odot \mathbf{m}^\mathrm{t}_j)_{w,h}, \tag{9}$$

where $j$ denotes the specified semantics, $e.g.$, neck, arms. To ensure that only $\mathbf{g}_\mathrm{un}$ with a size absolutely larger than $\mathbf{g}$ participates in the training of the stage represented by Eq. (7), we introduce a gating coefficient $\gamma$. Following [6], when the arm and neck sizes of $\mathbf{p}_\mathrm{un}$ are both smaller than those of $\mathbf{p}^\star$, we set $\gamma = 1$ (represents that backpropagation can be performed). In this case, $\mathbf{g}_\mathrm{un}$ can fully cover the original clothing of $\mathbf{p}^\star$, expressed as Eq. (10):

$$\gamma = \begin{cases} 1, & \text{if } \mathcal{A}(\mathbf{m}^\mathrm{s}_\mathrm{arm}, \mathbf{m}^\mathrm{t}_\mathrm{arm}) \geq 0 \text{ and } \mathcal{A}(\mathbf{m}^\mathrm{s}_\mathrm{neck}, \mathbf{m}^\mathrm{t}_\mathrm{neck}) \geq 0, \\ 0, & \text{otherwise.} \end{cases} \tag{10}$$

where $\mathbf{m}_\mathrm{arm} \in \{0,1\}^{H \times W}$ and $\mathbf{m}_\mathrm{neck} \in \{0,1\}^{H \times W}$ represent the semantic layers of the arms and the neck, respectively. By performing the occlusion elimination task represented in Eq. (7), the model can effectively learn to eliminate the acquired occlusions and achieve a favorable convergence. However, there exists a *distribution shift* between the pseudo reference image $\mathbf{p}_\mathrm{un}$ and the real data $\mathbf{p}$, which will bound the performance of $\mathcal{M}_\theta$.

**Correct Distribution Shift.** To correct the distribution shift, we perform the reconstruction task with a probability of $\eta = 10\%$ during training, thereby expanding the model's perception of the real distribution. This process is represented as Eq. (11):

$$\theta^* = \arg\min_\theta \ \mathcal{L}_\mathrm{dist}\big(\mathbf{p}, \ \mathcal{M}_\theta(\mathbf{p}, \mathbf{g})\big). \tag{11}$$

### 5.3 Plug-and-Play De-occlusion for Generative Models

There are now many types of generative networks, such as the classic GANs [25] and diffusion models [23]. *The de-occlusion approach we discussed above can be fully applied to both.* However, for diffusion models, due to their unique Markov process, the implementation process differs slightly from that of GANs. We implemented $\mathcal{M}_\theta$ using the diffusion model in this work, and the overall loss corresponding to Eqs. (7) and (11) is expressed as Eq. (12):

$$\min_\theta \begin{cases} \gamma \cdot \mathcal{L}_\mathrm{LDM}(\mathcal{M}_\theta(\mathbf{p}_\mathrm{un}, \mathbf{g}), \mathbf{p}^\star), & \text{if } p > \eta, \\ \mathcal{L}_\mathrm{LDM}(\mathcal{M}_\theta(\mathbf{p}, \mathbf{g}), \mathbf{p}), & \text{otherwise.} \end{cases} \tag{12}$$

where $p = \mathtt{random}(0,1)$ denotes the probability value generated randomly. Limited by space, specific implementation details are illustrated in **Technical Appendices** (Sec. B).

Table 2: **Quantitative comparisons on the VITON-HD and DressCode datasets.** For LPIPS, FID, and KID, the lower the better. For SSIM, the higher the better."Mask-Free" denotes whether the inpainting mask $m_{agn}$ and human structural representation (HSR) $r$ are used during *inference*. **Bold** denotes the best result. Underline represents second best.

| Train / Test Methods | Publication | Backbone | Mask-Free | VITON-HD | | | | DressCode Upper | | | |
|---|---|---|---|---|---|---|---|---|---|---|---|
| | | | | $SSIM_p \uparrow$ | $LPIPS_p \downarrow$ | $FID_{up} \downarrow$ | $KID_{up} \downarrow$ | $SSIM_p \uparrow$ | $LPIPS_p \downarrow$ | $FID_{up} \downarrow$ | $KID_{up} \downarrow$ |
| **VITON-HD** [21] | CVPR'21 | ResUnet | ✗ | 0.862 | 0.117 | 12.117 | 3.23 | n/a | n/a | n/a | n/a |
| **HR-VITON** [26] | ECCV'22 | ResUnet | ✗ | 0.878 | 0.105 | 11.265 | 2.73 | 0.936 | 0.065 | 13.82 | 2.71 |
| **GP-VTON** [4] | CVPR'23 | ResUnet | ✗ | 0.884 | 0.081 | 9.701 | 1.26 | 0.769 | 0.270 | 20.11 | 8.17 |
| **LaDI-VTON** [10] | MM'23 | SD1.5 | ✗ | 0.864 | 0.096 | 9.480 | 1.99 | 0.915 | 0.063 | 14.26 | 3.33 |
| **PbE** [11] | CVPR'23 | SD1.5 | ✗ | 0.802 | 0.143 | 11.939 | 3.85 | 0.897 | 0.078 | 15.33 | 4.64 |
| **DCI-VTON** [5] | MM'23 | SD1.5 | ✗ | 0.880 | 0.080 | 8.998 | 1.19 | **0.937** | **0.042** | 11.92 | 1.89 |
| **StableVITON** [7] | CVPR'24 | SD1.5 | ✗ | 0.864 | 0.084 | 9.465 | 1.40 | n/a | n/a | n/a | n/a |
| **StableGarment** [27] | arXiv'24 | SD1.5 | ✗ | 0.803 | 0.104 | 17.115 | 8.85 | n/a | n/a | n/a | n/a |
| **Anydoor** [13] | CVPR'24 | SD1.5 | ✗ | 0.821 | 0.099 | 10.850 | 2.46 | 0.899 | 0.119 | 14.83 | 3.05 |
| **IDM-VTON** [28] | ECCV'24 | SDXL | ✗ | 0.850 | 0.060 | 9.842 | 1.12 | 0.880 | 0.056 | 9.54 | 4.32 |
| **LDE-VTON** [16] | AAAI'25 | SD1.5 | ✗ | 0.884 | 0.081 | 9.640 | 1.21 | n/a | n/a | n/a | n/a |
| **CatVTON** [9] | ICLR'25 | SD1.5 | ✗ | 0.870 | 0.061 | 9.287 | 1.17 | 0.902 | 0.045 | 7.40 | 2.62 |
| **BooW-VTON** [14] | CVPR'25 | SDXL | ✓ | 0.862 | 0.108 | **8.809** | **0.82** | 0.919 | 0.062 | 11.03 | **0.86** |
| **Ours** | This Work | SD1.5 | ✓ | **0.889** | **0.057** | 8.854 | 0.96 | 0.923 | **0.042** | **6.58** | 1.72 |

- n/a: official code or data is inaccessible.

# 6 Experiments and Analysis

**Implementation.** Our model is built on the `Diffusers` framework[4] with Stable Diffusion v1.5 as the backbone and initialized from the official CatVTON checkpoint [9]. It is fine-tuned for 100 epochs on six NVIDIA RTX 4090 GPUs under Ubuntu 22.04 LTS. Training employs $T = 1,000$ denoising steps with a linear noise schedule, the AdamW [29] optimizer ($\beta_1 = 0.5$, $\beta_2 = 0.999$) in `fp32` precision, a batch size of 8, and a learning rate of $1 \times 10^{-5}$.

**Datasets and Metrics.** We conduct experiments on three challenging datasets: **VI-TON** [20], **VITON-HD** [21], and **Dress-Code** [22]. **VITON** dataset contains 16,253 image groups, each with a resolution of $256 \times 192$. It is divided into a training set of 14,221 groups and a testing set of 2,032 groups. **VITON-HD** dataset, with a resolution of $512 \times 384$, comprises 13,679 image groups and is split into a training set of 11,647 groups and a testing set of 2,032 groups. **DressCode** dataset, also with a resolution of $512 \times 384$, includes 15,363 image groups and is divided into a training set of 12,863 groups and a testing set of 2,500

Table 3: **Quantitative comparisons on the VITON dataset.**

| Methods | Publication | Mask-Free | $SSIM_p \uparrow$ | $FID_{up} \downarrow$ |
|---|---|---|---|---|
| **PF-AFN** [2] | CVPR'21 | ✓ | 0.89 | 10.21 |
| **RT-VTON** [30] | CVPR'22 | ✗ | n/a | 11.66 |
| **DAFlow** [31] | ECCV'22 | ✗ | 0.88 | 12.05 |
| **DressCode** [22] | CVPR'22 | ✗ | 0.89 | 13.71 |
| **POVNet** [32] | TPAMI'23 | ✗ | 0.89 | 13.37 |
| **USC-PFN** [15] | NeurIPS'23 | ✓ | 0.91 | 10.47 |
| **PbE** [11] | CVPR'23 | ✗ | 0.83 | 12.56 |
| **TPD** [12] | CVPR'24 | ✗ | 0.89 | 9.58 |
| **LDE-VTON** [16] | AAAI'25 | ✗ | 0.91 | 9.86 |
| **Ours** | **This Work** | ✓ | 0.91 | **9.23** |

- n/a: official code or data is inaccessible.

groups. All evaluations and visualizations are performed using the test set. For metrics, under paired setting $(p, g)$, we employ the **SSIM** (Structural Similarity Index Measure) [33] to assess the pixel-level similarity between the generated and ground-truth images. Meanwhile, we utilize **LPIPS** (Learned Perceptual Image Patch Similarity) [34] to measure the perceptual similarity, which captures the semantic and high-level features of the images. For evaluating the overall distribution of generated images compared to real images, under unpaired setting $(p, g_{un})$, we calculate the **FID** (Fréchet Inception Distance) [35] and **KID** (Kernel Inception Distance) [36], which provide insights into how well our method can produce images that match the real-world data distribution.

**Baselines.** We utilize 21 state-of-the-art (SOTA) methods for comprehensive evaluation.

• **GAN-based Methods**: PF-AFN [2], RT-VTON [30], DAFlow [31], DressCode [22], POVNet [32], USC-PFN [15], VITON-HD [21], HR-VITON [26], and GP-VTON [4].

---

[4] https://github.com/huggingface/diffusers

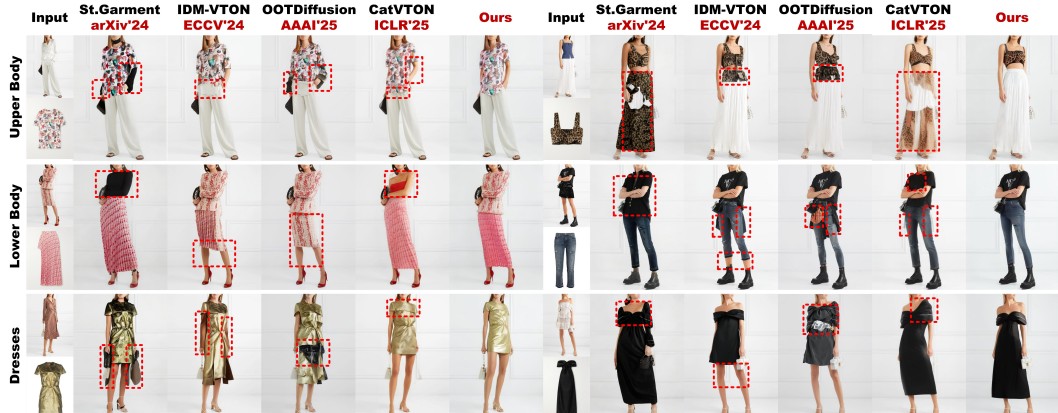

Figure 4: **Qualitative results** on the DressCode dataset. The baseline methods consist of four SOTA diffusion-based methods. Red dashed boxes highlight the limitations of each method.

• **Diffusion-based Methods**: LaDI-VTON [10], PbE [11], DCI-VTON [5], TPD [12], StableVITON [7], OOTDiffusion [8], AnyDoor [13], StableGarment [27], IDM-VTON [28], LDE-VTON [16], CatVTON [9], and BooW-VTON [14].

## 6.1 Comparison with Baseline Methods

We conducted a quantitative comparison of our method against 21 baseline methods, as shown in Tab. 3. On the low-resolution dataset VITON [20], our method outperforms existing technologies in terms of FID, while achieving comparable results to the SOTA methods in terms of SSIM. This is mainly because the residual parts of the source person are favorable for the SSIM metric (for reconstruction). As shown in Tab. 2, on the two high-resolution datasets VITON-HD [21] and DressCode [22], our method slightly underperforms the SOTA BooW-VTON in some metrics. One significant reason for this is that BooW-VTON utilizes a more powerful SDXL backbone [24].

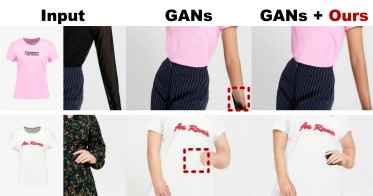

Figure 5: **Applying our training framework to GANs.**

Apart from this, our method outperforms existing technologies in most cases and also secures the highest overall ranking. This demonstrates that our method can effectively address occlusion issues while generalizing well to try-on scenarios with various clothing styles.

For a visual assessment[5], we present qualitative comparison on VITON-HD [21] and DressCode [22]. As shown in Figs. 3 and 4. It can be seen that regardless of the style of the dataset, our framework can handle both types of occlusions with ease, especially the significant inherent occlusions. Our method can almost completely eliminate this issue, which demonstrates the generalization and universality of the proposed approach. Furthermore, as shown in Fig. 5, we applied our training framework to GANs. The occlusion-free results obtained confirm that our framework can be generalized to any generative network type (GANs or diffusion models, *etc.*).

## 6.2 Ablation Studies

We study the effectiveness of each design choice in our framework and draw the following conclusions: (**#1**) **Correct Covering Process.** We replaced $g_{un}$ in Eq. (7) with both a randomly selected $g_{un}$ and one that had been filtered through our strategy, in order to verify the effectiveness of our filtering approach. As shown in Fig. 6 and Tab. 4, using a randomly selected $g_{un}$ still resulted in significant acquired occlusion. However, after applying our filtering strategy to the baseline (CatVTON [9]), the phenomenon of acquired occlusion was largely eliminated. This indicates that enhancing the model's ability to eliminate occlusions in covered samples can improve the model's capability to parse human structures. (**#2**) **Background Pre-Replacement.** Without background pre-replacement,

---

[5]More visualization results are shown in the **Technical Appendices** (Sec. D) and **Supplementary Material**

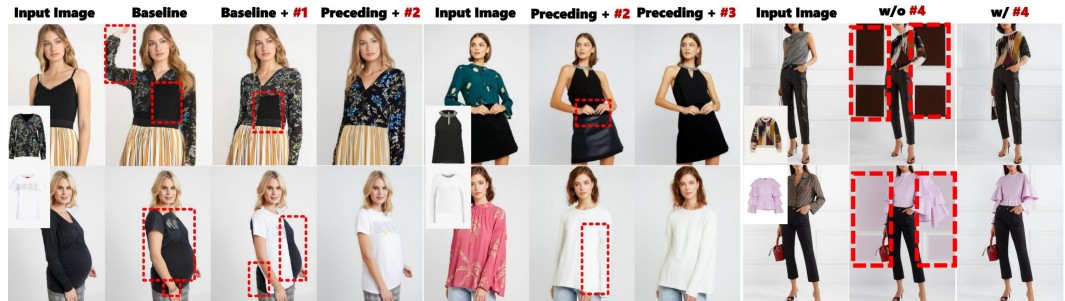

Figure 6: **Visual ablation studies** of different components in our approach. Zooming in for more details. Red dashed boxes highlight the limitations of each configuration.

the try-on results of configuration #2 in Fig. 6 exhibited significant inherent occlusion. However, after introducing $\mathbf{m}_{RB}$ to replace the background of $\mathbf{p}$, the inherent occlusion was eliminated, thereby demonstrating the effectiveness of the background pre-replacement. (**#3**) **Correct Distribution Shift.** We verified its effectiveness by separately adding and removing the correct distribution shift process. As can be seen from Fig. 6 and Tab. 4, when the correct distribution shift is removed, some samples exhibit synthetic bias, which is caused by the erroneous distribution of some defective $\mathbf{p}_{un}$ misleading the model training process. Therefore, when it is added, this part of the defect is significantly improved. (**#4**) **In-the-Wild Scenes.** Firstly, Tab. 4 presents our background replacement method for datasets with pure backgrounds. Additionally, we verified the generalization of replacing the background of $\mathbf{p}$ with $\mathbf{B}$ in real-world scenarios, *i.e.*, testing the model trained on the pure background VITON-HD dataset with the DressCode dataset. The results in Fig. 6 show that our method can better handle person images with diverse background styles, indicating that using the synthesized $\mathbf{B}$ can significantly enhance the model's ability to distinguish between the human body and background.

## 7 Conclusion

In this work, we present a novel mask-free VTON framework to address the inherent and acquired occlusion problems that plague existing VTON methods. Our framework introduces two key operations: the background pre-replacement operation and the covering-and-eliminating operation. The background pre-replacement operation effectively mitigates inherent occlusions by preventing the model from confusing target clothing information with the human body or image background. The covering-and-eliminating operation enhances the model's ability to understand and model human semantic structures, thereby reducing acquired occlusions. Our method is highly generalizable and can be easily integrated into various generative network architectures, such as GANs and diffusion models, in a plug-and-play manner. Extensive experiments on three VTON datasets validate the effectiveness and generalization ability of our proposed framework.

Table 4: **Ablation studies on VITON-HD.**

| Configuration | $\mathbf{SSIM}_p \uparrow$ | $\mathbf{LPIPS}_{up} \downarrow$ | $\mathbf{FID}_{up} \downarrow$ | $\mathbf{KID}_{up} \downarrow$ |
|---|---|---|---|---|
| Baseline | 0.870 | 0.061 | 9.287 | 1.17 |
| **#1** | 0.883 | 0.059 | 8.902 | 1.11 |
| **#2** | 0.882 | 0.060 | 8.863 | 1.04 |
| **#3** | 0.887 | 0.058 | 8.859 | 0.98 |
| **#4** | **0.889** | **0.057** | **8.854** | **0.96** |

## Acknowledgments

This work was in part supported by the National Key Research and Development Program of China (Grant No. 2022ZD0160604), the National Natural Science Foundation of China (NSFC, Grant No. 62176194), the Key Research and Development Program of Hubei Province (Grant No. 2023BAB083), the Project of Sanya Yazhou Bay Science and Technology City (Grant Nos. SCKJ-JYRC-2022-76, SKJC-2022-PTDX-031), the Project of Sanya Science and Education Innovation Park of Wuhan University of Technology (Grant No. 2021KF0031), and the Huawei Kunpeng-Ascend Innovation Incentive Programme. This work was also supported in part by both computing resources from the Wuhan Supercomputing Center and the Wuhan Artificial Intelligence Computing Center.

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

# Appendix

For a thorough understanding and visualization of our proposed framework, we compile a comprehensive appendix.

## A  Preliminary: Diffusion Models

Diffusion models [37, 38, 39], as probabilistic generative models, encompass a two-step process: diffusion and its reverse. The diffusion phase adheres to a Markov chain defined by $q(\boldsymbol{z}_t | \boldsymbol{z}_{t-1}) = \mathcal{N}(\boldsymbol{z}_t; \sqrt{1 - \beta_t} \boldsymbol{z}_{t-1}, \beta_t \mathbf{I})$, spanning $T$ iterations with a noise schedule $\{\beta_t\}_{t=1}^{T}$. This schedule incrementally corrupts the initial data, $\boldsymbol{z}_0 \sim q(\boldsymbol{z}_0)$, with Gaussian noise. Each noisy latent state $\boldsymbol{z}_t$ at any timestep $t$ can be sampled directly through a closed-form sampling function:

$$\boldsymbol{z}_t := \sqrt{\bar{\alpha}_t} \boldsymbol{z}_0 + \sqrt{1 - \bar{\alpha}_t} \boldsymbol{\epsilon}, \ \boldsymbol{\epsilon} \sim \mathcal{N}(0, \mathbf{I}), \tag{13}$$

where $t$ is uniformly sampled from $\{1, \ldots, T\}$. The noise level is determined by $\alpha_t = 1 - \beta_t$, and $\bar{\alpha}_t = \prod_{s=1}^{t} \alpha_s$. The reverse process starts with a noisy data $\boldsymbol{z}_T \sim \mathcal{N}(0, \mathbf{I})$ at step $T$ and gradually denoises it using known real distributions $q(\boldsymbol{z}_{t-1} | \boldsymbol{z}_t)$ for each step:

$$p_\theta(\boldsymbol{z}_{t-1} | \boldsymbol{z}_t) = \mathcal{N}(\boldsymbol{z}_{t-1}; \mu_\theta(\boldsymbol{z}_t, t), \Sigma_\theta(\boldsymbol{z}_t, t)). \tag{14}$$

To achieve this, a denoising autoencoder $\epsilon_\theta(\cdot)$ is trained to remove noise $\epsilon$ from $\boldsymbol{z}_t$ to reconstruct $\boldsymbol{z}_0$ by optimizing the following objective:

$$\min_\theta \ \mathbb{E}_{\boldsymbol{z}_0, \boldsymbol{\epsilon}, t} \| \boldsymbol{\epsilon}_\theta(\boldsymbol{z}_t, t) - \boldsymbol{\epsilon} \|_2^2. \tag{15}$$

## B  Training and Inference Procedures

The training and inference procedures of our proposed VTON framework are designed to address the inherent and acquired occlusion issues while ensuring high-quality occlusion-free virtual try-on results. The detailed procedures are outlined in Algorithm 1. The training process iterates until convergence, optimizing the model parameters $\theta$ to achieve high-quality occlusion-free virtual try-on. Additionally, for diffusion models, our training objective for Eq. (7) is formulated as follows:

$$\min_\theta \ \mathbb{E}_{\mathbf{z}_0, \boldsymbol{\epsilon}_\mathbf{x} \sim \mathcal{N}(0, \mathbf{I}), t \sim \mathcal{U}(1, T)} \left[ \left\| \boldsymbol{\epsilon}_\mathbf{x} - \boldsymbol{\epsilon}_{\boldsymbol{\theta}}(\mathbf{z}_t, \mathrm{Cat}_S(\mathbf{z}_{\mathbf{p}_{\mathrm{un}}}, \mathbf{z}_\mathbf{g}), t) \right\|_2^2 \right], \tag{16}$$

where

$$\mathbf{z}_t = \sqrt{\bar{\alpha}_t} \boldsymbol{\epsilon}_\mathbf{x} - \sqrt{1 - \bar{\alpha}_t} \mathrm{Cat}_S(\mathbf{z}_{\mathbf{p}^\star}, \mathbf{z}_\mathbf{g}), \ \boldsymbol{\epsilon}_\mathbf{x} \sim \mathcal{N}(0, \mathbf{I}) \in \mathbb{R}^{4 \times \frac{H}{8} \times \frac{W}{4}}, \tag{17}$$

$\mathbf{z}_{\mathbf{p}_{\mathrm{un}}} \in \mathbb{R}^{4 \times \frac{H}{8} \times \frac{W}{8}} = \mathcal{E}_{VAE}(\mathbf{p}_{\mathrm{un}})$, $\mathbf{z}_\mathbf{g} \in \mathbb{R}^{4 \times \frac{H}{8} \times \frac{W}{8}} = \mathcal{E}_{VAE}(\mathbf{g})$, $\mathbf{z}_{\mathbf{p}^\star} \in \mathbb{R}^{4 \times \frac{H}{8} \times \frac{W}{8}} = \mathcal{E}_{VAE}(\mathbf{p}^\star)$. $\mathrm{Cat}_S(\cdot)$ denotes the concatenation operation along the **spatial** dimension. $\mathcal{E}$ is the encoder of KL-regularized autoencoder with its default latent-space downsampling factor $f = 8$. In addition, our training objective for Eq. (11) is formulated as follows:

$$\min_\theta \ \mathbb{E}_{\mathbf{z}_0, \boldsymbol{\epsilon}_\mathbf{x} \sim \mathcal{N}(0, \mathbf{I}), t \sim \mathcal{U}(1, T)} \left[ \left\| \boldsymbol{\epsilon}_\mathbf{x} - \boldsymbol{\epsilon}_{\boldsymbol{\theta}}(\mathbf{z}_t, \mathrm{Cat}_S(\mathbf{z}_\mathbf{p}, \mathbf{z}_\mathbf{g}), t) \right\|_2^2 \right], \tag{18}$$

where

$$\mathbf{z}_t = \sqrt{\bar{\alpha}_t} \boldsymbol{\epsilon}_\mathbf{x} - \sqrt{1 - \bar{\alpha}_t} \mathrm{Cat}_S(\mathbf{z}_\mathbf{p}, \mathbf{z}_\mathbf{g}), \ \boldsymbol{\epsilon}_\mathbf{x} \sim \mathcal{N}(0, \mathbf{I}) \in \mathbb{R}^{4 \times \frac{H}{8} \times \frac{W}{4}}, \tag{19}$$

$\mathbf{z}_\mathbf{p} \in \mathbb{R}^{4 \times \frac{H}{8} \times \frac{W}{8}} = \mathcal{E}_{VAE}(\mathbf{p})$.

During the inference phase, the pre-trained well-performed try-on model $\mathcal{M}_\theta^*$ is used to generate virtual try-on results.

---

**Algorithm 1:** Training and Inference Procedures

---

**Input:** Dataset $\mathbf{D}_{\mathrm{real}}$, pre-trained T2I model, pre-trained parser $S$, pre-trained model $\mathcal{M}_\psi^*$
**Output:** Trained model $\mathcal{M}_\theta^*$

1  Training Procedure:
2  **repeat**
3     Obtain the input sample:
4       $(\mathbf{g}, \mathbf{g}_{\mathrm{un}}, \mathbf{p}) \sim \mathbf{D}_{\mathrm{real}}$ // Target clothing image, random clothing image, and reference person image from the dataset
5     Obtain the segmentation masks:
6       $\mathbf{m}_{\mathrm{agn}} \sim \mathbf{D}_{\mathrm{real}}$ // inpainting mask from the dataset
7       $\mathbf{m}_{\mathrm{ide}} \sim \mathbf{D}_{\mathrm{real}}$ // Identity layers (e.g., head, hands, feet) from the dataset
8     Generate the background-replaced person image:
9       $\mathbf{m}_{\mathrm{RB}} \leftarrow 1 - (\mathbf{m}_{\mathrm{agn}} + \mathbf{m}_{\mathrm{ide}} \odot (1 - \mathbf{m}_{\mathrm{agn}}))$; // Compute the residual background mask.
10      $\mathbf{B} \leftarrow \mathrm{T2I}(c, \epsilon)$; // Generate a random background image using a pre-trained T2I model.
11      $\mathbf{p}^\star \leftarrow \mathbf{m}_{\mathrm{RB}} \odot \mathbf{p} + (1 - \mathbf{m}_{\mathrm{RB}}) \odot \mathbf{B}$; // Replace the background.
12    Check the size of the clothing:
13      $\mathbf{m}_{\mathrm{t}} \leftarrow S(\mathbf{p}^\star)$; // Generate the human parsing map of the person image using a pre-trained parser $S$.
14      $\gamma \leftarrow \begin{cases} 1, & \text{if } \mathcal{A}(\mathbf{m}_{\mathrm{s}}^{\mathrm{neck}}, \mathbf{m}_{\mathrm{t}}^{\mathrm{neck}}) \geq 0 \text{ and } \mathcal{A}(\mathbf{m}_{\mathrm{s}}^{\mathrm{arm}}, \mathbf{m}_{\mathrm{t}}^{\mathrm{arm}}) \geq 0, \\ 0, & \text{otherwise.} \end{cases}$; // Check if the clothing can fully cover the original one.
15    Compute the pseudo reference image:
16      $\mathbf{p}_{\mathrm{un}} \leftarrow \mathcal{M}_\psi^*(\mathbf{p}^\star, \mathbf{g}_{\mathrm{un}})$; // Generate the pseudo reference image using the teacher model.
17    Perform the de-occlusion task:
18    **if** $\gamma = 1$ **then**
19       $\nabla_\theta \mathcal{L}_{\mathrm{DM}}(\mathcal{M}_\theta(\mathbf{p}_{\mathrm{un}}, \mathbf{g}), \mathbf{p}^\star)$; // Minimize the loss between the generated image and the pseudo reference image.
20    **end**
21    Correct the distribution shift:
22      $p \sim \mathrm{Uniform}(0, 1)$; // Generate a random probability value.
23    **if** $p < \eta$ **then**
24       $\nabla_\theta \mathcal{L}_{\mathrm{DM}}(\mathcal{M}_\theta(\mathbf{p}, \mathbf{g}), \mathbf{p})$; // Minimize the loss between the generated image and the real image.
25    **end**
26 **until** *converged*;
27 Inference Procedure:
   **Input:** Target clothing image $\mathbf{g}_{\mathrm{un}}$, reference person image $\mathbf{p}$
   **Output:** Virtual try-on result $\mathbf{p}_{\mathrm{try}}$
28 Generate the virtual try-on result:
29   $\mathbf{p}_{\mathrm{try}} \leftarrow \mathcal{M}_\theta^*(\mathbf{p}, \mathbf{g}_{\mathrm{un}})$; // Generate the virtual try-on result.
30 **return** $\mathbf{p}_{\mathrm{try}}$

---

## C  Technical Appendices

### C.1  Improve Traditional Mask-Free Method

For traditional mask-free methods, such as WUTON [1], PF-AFN [2], and LDE-VTON [16], solving inherent occlusions can be achieved with just a few simple steps. First, we generate teacher knowledge through a pre-trained well-performed teacher (inpainting) model $\mathcal{M}_\phi^*$:

$$\mathbf{t}_{\mathrm{un}} = \mathcal{M}_\phi^*\big(\mathbf{m}_{\mathrm{agn}}, \mathbf{p}_{\mathrm{agn}}, \mathbf{g}_{\mathrm{un}}, \mathbf{r}\big). \tag{20}$$

Then, we use $\mathbf{m}_{\mathrm{RB}}$ to perform *background pre-replacement* on $\mathbf{t}_{\mathrm{un}}$:

$$\mathbf{t}_{\mathrm{un}}^{\star} = \mathbf{m}_{\mathrm{RB}} \odot \mathbf{B} + (1 - \mathbf{m}_{\mathrm{RB}}) \odot \mathbf{t}_{\mathrm{un}}. \tag{21}$$

Finally, $\mathbf{t}_{\mathrm{un}}^{\star}$ is paired with $\mathbf{p}^{\star}$, where $\mathbf{p}^{\star}$ can be used as the input, while $\mathbf{t}_{\mathrm{un}}^{\star}$ serves as the pseudo reference image. The distillation process is then carried out by training a student network $\mathcal{M}_{\psi}$:

$$\psi^{*} = \arg\min_{\psi} \ \mathcal{L}_{\mathrm{dist}}\left(\mathbf{t}_{\mathrm{un}}^{\star}, \mathcal{M}_{\psi}(\mathbf{p}^{\star}, \mathbf{g}_{\mathrm{un}})\right). \tag{22}$$

Alternatively, $\mathbf{t}_{\mathrm{un}}^{\star}$ can be used as the input, while $\mathbf{p}^{\star}$ is used as the ground truth to train the student network $\mathcal{M}_{\psi}$:

$$\psi^{*} = \arg\min_{\psi} \ \mathcal{L}_{\mathrm{dist}}\left(\mathbf{p}^{\star}, \mathcal{M}_{\psi}(\mathbf{t}_{\mathrm{un}}^{\star}, \mathbf{g})\right). \tag{23}$$

### C.2 Size Comparison for Other Types of Clothing

To ensure that only $\mathbf{g}_{\mathrm{un}}$ with a size absolutely larger than $\mathbf{g}$ participates in the training of the stage represented by Eq. (7), in addition to the execution process for upper clothing described in the main text, the following are the execution processes for the remaining two types of clothing.

**Lower Clothing.** For lower clothing, when the leg size of $\mathbf{p}_{\mathrm{un}}$ is smaller than those of $\mathbf{p}^{\star}$, we set $\gamma = 1$. In this case, $\mathbf{g}_{\mathrm{un}}$ can fully cover the original lower clothing of $\mathbf{p}^{\star}$, expressed as Eq. (24):

$$\gamma = \begin{cases} 1, & \text{if } \mathcal{A}(\mathbf{m}_{\mathrm{leg}}^{\mathrm{s}}, \mathbf{m}_{\mathrm{leg}}^{\mathrm{t}}) \geq 0, \\ 0, & \text{otherwise,} \end{cases} \tag{24}$$

where $\mathbf{m}_{\mathrm{leg}} \in \{0,1\}^{H \times W}$ represents the semantic layer of the legs. For the person identity mask $\mathbf{m}_{\mathrm{ide}}$, we obtained the layers "hair," "shoes," "hat," "sunglasses," "scarf," "bag," "head," "upper_clothes," and "arms" from human parsing as the sub-layers that compose $\mathbf{m}_{\mathrm{ide}}$. The code is represented as:

```
m_ide = (parse_array == label_map["hair"]).astype(np.float32) + \
    (parse_array == label_map["left_shoe"]).astype(np.float32) + \
    (parse_array == label_map["right_shoe"]).astype(np.float32) + \
    (parse_array == label_map["hat"]).astype(np.float32) + \
    (parse_array == label_map["sunglasses"]).astype(np.float32) + \
    (parse_array == label_map["scarf"]).astype(np.float32) + \
    (parse_array == label_map["bag"]).astype(np.float32) + \
    (parse_array == label_map["head"]).astype(np.float32) + \
    (parse_array == label_map["upper_clothes"]).astype(np.float32) + \
    (parse_array == label_map["left_arm"]).astype(np.float32) + \
    (parse_array == label_map["right_arm"]).astype(np.float32)
```

**Dresses.** For dresses, when the arm, neck, and leg sizes of $\mathbf{p}_{\mathrm{un}}$ are smaller than those of $\mathbf{p}^{\star}$, we set $\gamma = 1$. In this case, $\mathbf{g}_{\mathrm{un}}$ can fully cover the original dresses of $\mathbf{p}^{\star}$, expressed as Eq. (25):

$$\gamma = \begin{cases} 1, & \text{if } \mathcal{A}(\mathbf{m}_{\mathrm{neck}}^{\mathrm{s}}, \mathbf{m}_{\mathrm{neck}}^{\mathrm{t}}) \geq 0 \text{ and } \mathcal{A}(\mathbf{m}_{\mathrm{arm}}^{\mathrm{s}}, \mathbf{m}_{\mathrm{arm}}^{\mathrm{t}}) \geq 0 \text{ and } \mathcal{A}(\mathbf{m}_{\mathrm{leg}}^{\mathrm{s}}, \mathbf{m}_{\mathrm{leg}}^{\mathrm{t}}) \geq 0, \\ 0, & \text{otherwise.} \end{cases} \tag{25}$$

For $\mathbf{m}_{\mathrm{ide}}$, we obtained the layers "hair," "shoes," "hat," "sunglasses," "scarf," "bag," and "head," from human parsing as the sub-layers that compose $\mathbf{m}_{\mathrm{ide}}$. The code is represented as:

```
m_ide = (parse_array == label_map["hair"]).astype(np.float32) + \
    (parse_array == label_map["left_shoe"]).astype(np.float32) + \
    (parse_array == label_map["right_shoe"]).astype(np.float32) + \
    (parse_array == label_map["hat"]).astype(np.float32) + \
    (parse_array == label_map["sunglasses"]).astype(np.float32) + \
    (parse_array == label_map["scarf"]).astype(np.float32) + \
    (parse_array == label_map["bag"]).astype(np.float32) + \
    (parse_array == label_map["head"]).astype(np.float32)
```

**Upper Clothing.** For $\mathbf{m}_{\mathrm{ide}}$, we obtained the layers "hair," "shoes," "hat," "sunglasses," "scarf," "bag," "head," "skirt," "pants," "dress" "belt" and "legs" from human parsing as the sub-layers that compose $\mathbf{m}_{\mathrm{ide}}$. The code is represented as:

```
m_ide = m_ide + # m_ide of dresses
    (parse_array == label_map["skirt"]).astype(np.float32) + \
    (parse_array == label_map["pants"]).astype(np.float32) + \
    (parse_array == label_map["dress"]).astype(np.float32) + \
    (parse_array == label_map["belt"]).astype(np.float32) + \
    (parse_array == label_map["left_leg"]).astype(np.float32) + \
    (parse_array == label_map["right_leg"]).astype(np.float32)
```

### C.3  How to obtain B in Eq. (4)?

The background map $\mathbf{B}$ can be designed to be random scene images generated by a pre-trained well-performed T2I (Text-to-Image) model [23, 24]: $\mathbf{B} = \texttt{T2I}(\mathbf{c}, \epsilon)$, where $\mathbf{c}$ is the prompt and $\epsilon$ is random Gaussian noise. In our implementation, T2I utilizes the pre-trained SD1.5 implemented by the diffusers library in Huggingface[6]. By simply inputting a prompt $\mathbf{c}$ related to any scene, the desired scene image $\mathbf{B}$ can be obtained.

Alternatively, $\mathbf{B}$ can be designed as random single value matrix: $\mathbf{B} = \texttt{random}(0, 1) \in \mathbb{R}^{3 \times H \times W}$. This approach is simpler and more practical because we only require the model to focus on the human body in the image, without needing the model to understand what the background in the image is. Its code is implemented as follows:

```
mRB = 1 - (agnostic_mask + m_ide * (1-agnostic_mask))
B = torch.rand(1) # A random floating-point number between 0 and 1
image = (1 - mRB) * image + mRB * B
```

The reason for choosing a random floating-point number between 0 and 1 is that the normalized human image's original background color values generally range from 0 to 1, which allows $\mathbf{B}$ to better adapt to real sample distributions.

## D  More Results

We first supplement some quantitative results of classic methods on the VITON dataset [20] in Tab. 3, as shown in Table 5. In addition, we qualitatively compare our proposed method with several state-of-the-art (SOTA) methods.

### D.1  Results of VITON-HD

We conducted qualitative comparisons between our method and baseline methods on the VITON-HD dataset [21], with a resolution of $512 \times 384$. Figs. 7 and 8 illustrate the visual comparison of try-on results between our method, DCI-VTON [5], StableVITON [7], IDM-VTON [28], StableGarment [27], LDE-VTON [16], and CatVTON [9]. It is evident that our method can handle occlusion problem of higher-resolution images in different methods, resulting in more realistic try-on results.

Table 5: **Supplementary quantitative results of early virtual try-on methods on the VITON dataset in Tab. 3.** "Mask-Free" denotes whether the mask and structural representation are used during inference. **Bold** denotes the best result. Underline represents second best.

| Methods | Publication | Mask-Free | $\mathrm{SSIM}_p \uparrow$ | $\mathrm{FID}_{up} \downarrow$ |
|---|---|---|---|---|
| **VITON** [20] | CVPR'18 | ✗ | 0.74 | 55.71 |
| **CP-VTON** [40] | ECCV'18 | ✗ | 0.72 | 24.45 |
| **VTNFP** [41] | ICCV'19 | ✗ | 0.80 | n/a |
| **Cloth-flow** [42] | CVPR'19 | ✗ | 0.84 | 14.43 |
| **CP-VTON+** [43] | CVPRW'20 | ✗ | 0.75 | 21.04 |
| **SieveNet** [44] | WACV'20 | ✗ | 0.77 | n/a |
| **ACGPN** [45] | CVPR'20 | ✗ | 0.84 | 16.64 |
| **LM-VTON** [46] | AAAI'21 | ✗ | 0.85 | 17.18 |
| **DCTON** [47] | CVPR'21 | ✗ | 0.83 | 14.82 |
| **ZFlow** [48] | ICCV'21 | ✗ | 0.88 | 15.17 |
| **OVNet** [49] | CVPR'21 | ✗ | 0.85 | 15.78 |
| **Ours** | **This Work** | ✓ | **0.91** | **9.23** |

- n/a: official code or data is inaccessible.

### D.2  Results of DressCode

Furthermore, we have also included additional visual comparisons between our method and baseline methods (IDM-VTON [28], StableGarment [27], and CatVTON [9]) on DressCode dataset [22] to

---

[6]https://huggingface.co/docs/diffusers/training/text2image

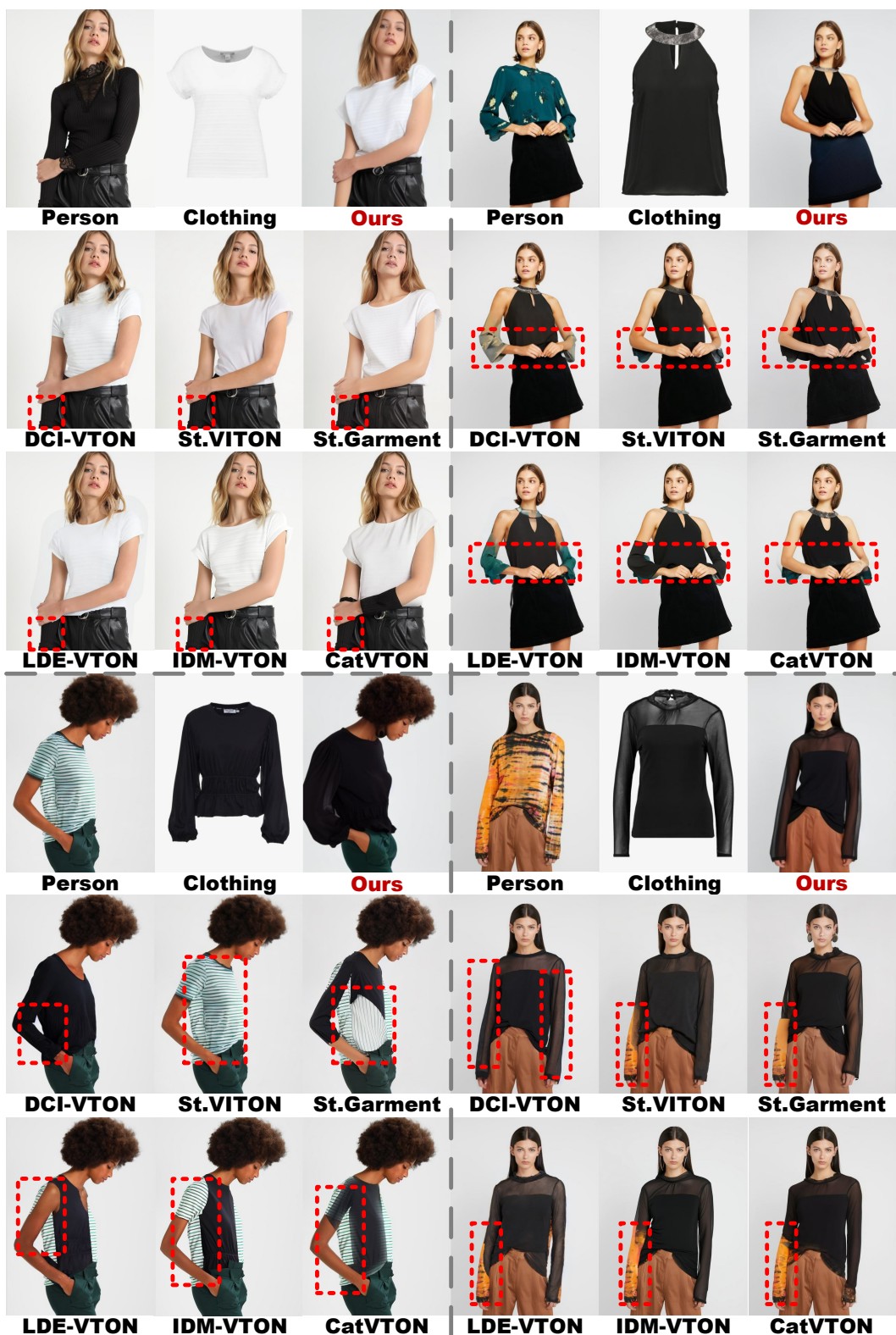

Figure 7: **Qualitative results** on the VITON-HD dataset. The baseline methods consist of six SOTA diffusion-based methods. Red dashed boxes highlight the limitations of each method.

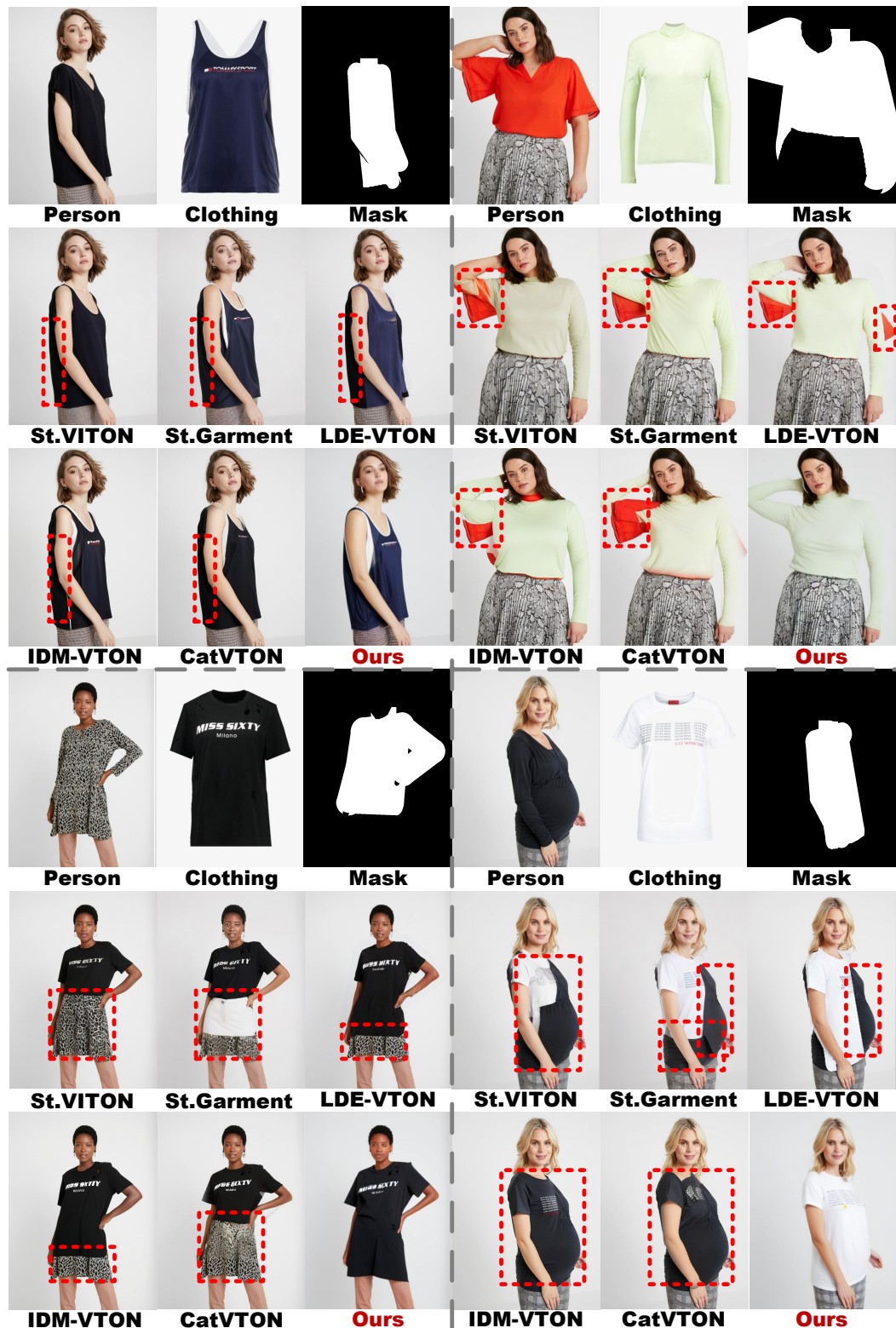

Figure 8: **Qualitative results** on the VITON-HD dataset. The baseline methods consist of five SOTA diffusion-based methods. Red dashed boxes highlight the limitations of each method.

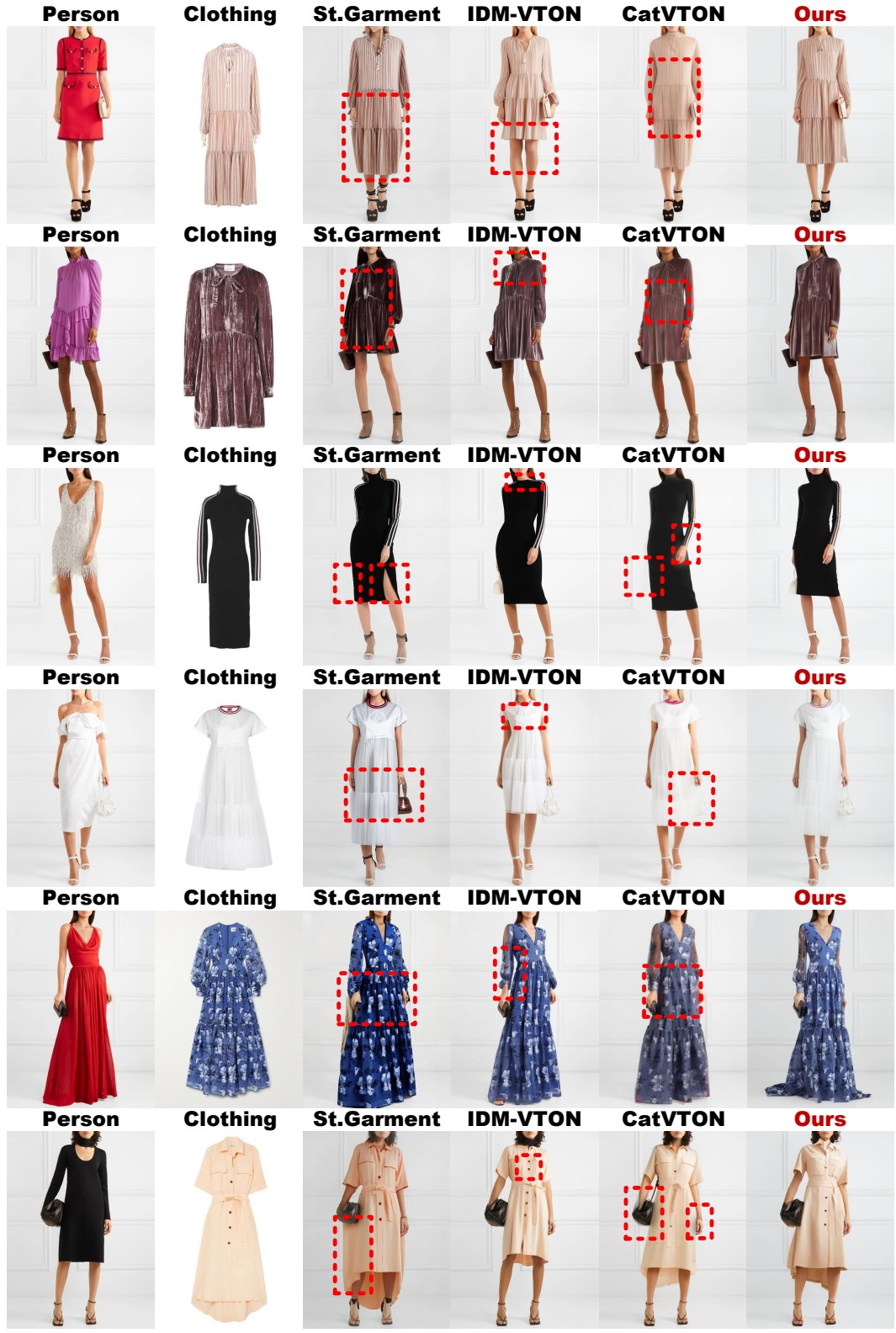

Figure 9: **Qualitative results** on the DressCode dataset (**Dresses**). The baseline methods consist of three SOTA diffusion-based methods. Red dashed boxes highlight the limitations of each method.

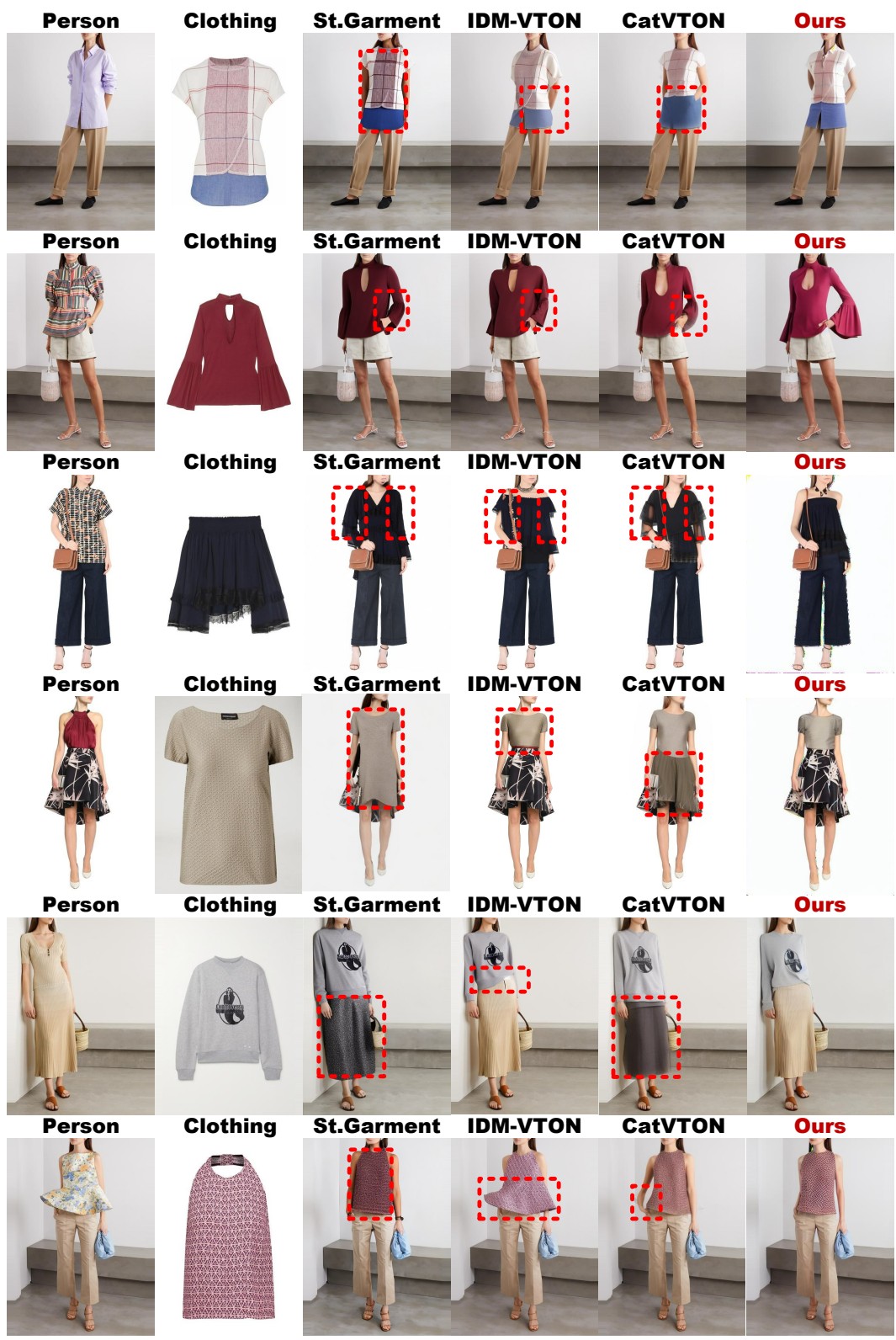

Figure 10: **Qualitative results** on the DressCode dataset (**Upper**). The baseline methods consist of three SOTA diffusion-based methods. Red dashed boxes highlight the limitations of each method.

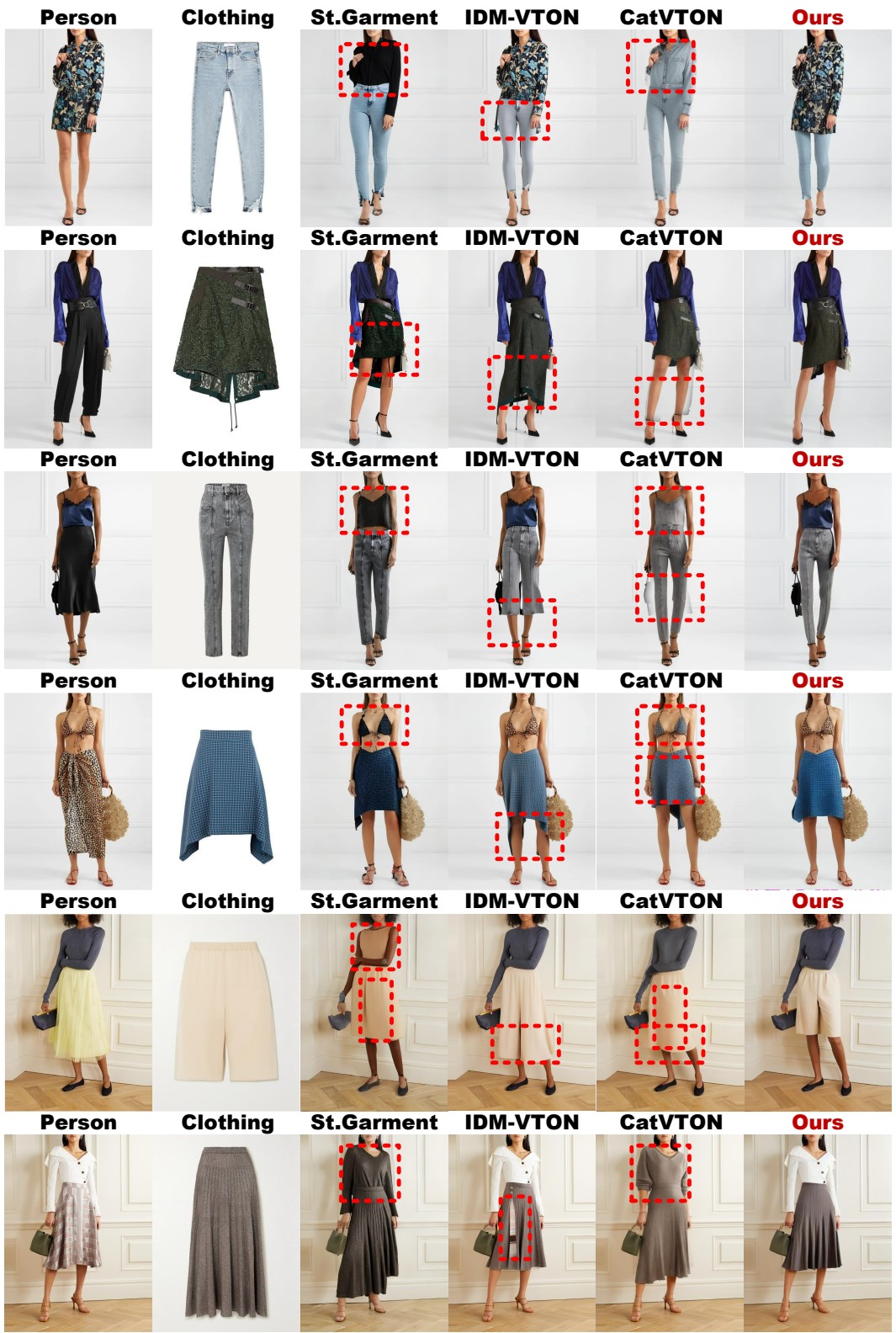

Figure 11: **Qualitative results** on the DressCode dataset (**Lower**). The baseline methods consist of three SOTA diffusion-based methods. Red dashed boxes highlight the limitations of each method.

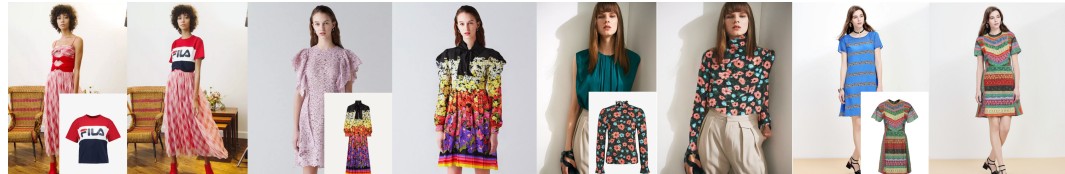

Figure 12: **Qualitative results** in in-the-wild scenarios.

demonstrate its superiority, as shown in Figs. 9 (Dresses), 10 (Upper Clothing), and 11 (Lower Clothing).

### D.3 In-the-Wild Results

To validate the robustness and generalization of the model in real-world, open-domain scenarios, we tested a set of in-the-wild samples. In the actual implementation, we migrated our de-occlusion architecture to [8] to leverage the powerful garment-image understanding capability of ReferenceNet. After fine-tuning, the visualized results are shown in Fig. 12.

## E Limitation and Future Work

Despite significant advancements in addressing occlusion issues for virtual try-on, this work still faces limitations, particularly regarding dataset quality. Current datasets, whether synthetic or real, may fail to capture the full range of human appearances, clothing styles, and occlusion scenarios, thus limiting model performance. Furthermore, our method may yield failed results when encountering clothing shapes that are objectively unlearnable, as shown in Fig. 13. In the currently available datasets, the lower body area indicated by the red dashed box in the swimsuit is covered by pants after trying it on. However, in real life, people generally do not wear other pants or skirts when wearing this kind of swimsuit. Due to the



Figure 13: **Example of a failed virtual try-on of our method.**

inconsistency between this ground truth and real-life situations, biases have been introduced in the try-on process for different clothing samples. The model cannot learn to synthesize this area, resulting in a failed try-on of the lower body area. However, this limitation is due to the lack of diversity in the dataset, rather than a flaw in the framework we designed. We believe that this issue can be resolved in the future by using a much larger number of training samples. Future work should focus on enriching dataset diversity and mitigating biases to enhance model robustness and applicability.

## F Discussion of Societal Impacts

The societal impacts of virtual try-on technology are multifaceted, offering significant benefits in terms of enhanced shopping experiences, fashion innovation, and environmental sustainability. However, these advancements also bring challenges related to privacy, digital divide, ethical considerations, and economic disruption. Addressing these challenges requires a collaborative effort from technologists, policymakers, and stakeholders across the retail and fashion industries to ensure that the benefits of virtual try-on technology are realized equitably and responsibly.

By engaging in ongoing discussions and implementing appropriate measures, we can harness the potential of virtual try-on technology to create a more inclusive, convenient, and sustainable future for fashion and retail.

