# OpenReview forum: "Mitigating Occlusions in Virtual Try-On via A Simple-Yet-Effective Mask-Free Framework"
_NeurIPS.cc/2025/Conference — NeurIPS 2025 poster_

### Official Review · Reviewer_ciwU · 2025-06-17

**Clarity:** 3
**Significance:** 3
**Originality:** 3
**Rating:** 4
**Confidence:** 4

**Summary:**

This work introduces a mask-free VTON framework that tackles occlusions through two key operations: background pre-replacement to reduce inherent confusion between clothing and background, and covering-and-eliminating to address acquired occlusions by enhancing semantic understanding. The method is generalizable and compatible with generative models like GANs and diffusion models.

**Questions:**

1) Do g and g_un both refer to upper cloth？
2) It appears that the identity of the subject changes noticeably after applying the target clothing. What might be the underlying cause of this issue? Have the authors considered any strategies to preserve identity more effectively during the try-on process?
3) Does the proposed method generalize well to in-the-wild images, particularly for subjects with greater variation in body shape, such as individuals with heavier or more muscular physiques?

**Ethical Concerns:**

["NO or VERY MINOR ethics concerns only"]

**Final Justification:**

Overall this is a nice work which advanced the virtual try on framework. The author has referred to existing figure for virtual try on results which are closed to in-the-wild cases. Although the effectiveness can be demonstrated to some extend, there is still a domain gap between fashion image taken in studio with set camera and daily image taken with random lenses, thus a more comprehensive evaluation on in the wild cases would be better. Although the author has explain why they did not include the inpainting method to address the identity changing issue, leaving this problem unsolved is still considered a limitation for the proposed work. I will keep my score unchanged.

**Limitations:**

Yes

**Quality:**

3

**Strengths And Weaknesses:**

Strengths:
1) The paper provides an effective solution to the primary occlusion issues commonly encountered in virtual try-on approaches, addressing both inherent and acquired occlusions
2) The method is extensively compared with state-of-the-art methods, clearly demonstrating its effectiveness and generalizability.


weaknesses:
1) The methodological descriptions and accompanying notations in the paper are a bit hard to follow. In particular, the explanation of the background pre-replacement strategy could be improved by supplementing the mathematical formulation with a clear, step-by-step visual illustration. Such a figure would enhance clarity and accessibility, as the current presentation may hinder comprehension.

2) The qualitative results are primarily demonstrated using images from fashion datasets or under controlled studio photography settings. It would be more meaningful to evaluate the proposed method on in-the-wild data to better assess its robustness and generalization to real-world scenarios.

---

> ### Author Rebuttal · Authors · 2025-07-31
>
> We appreciate all your valuable suggestions and constructive comments.
>
> > **W1**: The methodological descriptions and accompanying notations in the paper are a bit hard to follow. In particular, the explanation of the background pre-replacement strategy could be improved by supplementing the mathematical formulation with a clear, step-by-step visual illustration. Such a figure would enhance clarity and accessibility, as the current presentation may hinder comprehension.
> >> **R1**: We appreciate your valuable suggestions, they are very helpful in improving our paper. In the revised version of this paper, we will follow your comments to add the visual illustrations to supplement the mathematical formulation of each step in our designed background pre-replacement strategy, so as to enhance the clarity and accessibility of such operation.
>
> > **W2**: The qualitative results are primarily demonstrated using images from fashion datasets or under controlled studio photography settings. It would be more meaningful to evaluate the proposed method on in-the-wild data to better assess its robustness and generalization to real-world scenarios. **And Q3**: Does the proposed method generalize well to in-the-wild images, particularly for subjects with greater variation in body shape, such as individuals with heavier or more muscular physiques?
> >> **R2**: Due to the background augmentations used in the background pre-replacement operation, our method should technically be able to generalize to real-world in-the-wild images. In the first and last two rows of Figure 10 and the last two rows in Figure 11, we present some qualitative results with more practical backgrounds rather than plain colors. And moreover, the right block of Figure 8 also displays some results of heavier persons with sideways poses. Nevertheless, as suggested, we will add more results under in-the-wild scenarios in our revised version to further validate the generalization ability of our method.
>
> > **Q1**: Do g and g_un both refer to upper cloth？
> >> **RQ1**: Actually, $g$ and $g_{un}$ indicate different garments from the same cloth category (e.g., lower body, upper body, or dresses). Specifically, $g$ denotes the garment that matches the one worn by the person in image $p$, forming a paired sample, whereas $g_{un}$ is a garment randomly sampled from the dataset.
>
> > **Q2**: It appears that the identity of the subject changes noticeably after applying the target clothing. What might be the underlying cause of this issue? Have the authors considered any strategies to preserve identity more effectively during the try-on process?
> >> **RQ2**:  This is mainly due to the limitations of latent diffusion models. Encoding and decoding human images with a pre-trained KL-regularized autoencoder (e.g., VQGAN) in latent diffusion models will inevitably discard a certain amount of face details (especially around the eyes) [1]. Most existing virtual try-on methods often mitigate such information loss by performing the following post-processing step, which adopts an inpainting mask to paste the human identity area from the original image onto the same region of the generated result.
> >>
> >>$$p_{result} = m_{agn} \odot p + (1 - m_{agn}) \odot p_{un}$$
> >>However, since the inpainting mask used above may also be imprecise and introduce some new occlusions, we abandon this post-processing step in our proposed framework. Prompted by your comment, we will incorporate optimization of the latent diffusion model generation quality [1] in our future work to achieve further improvements.
> >>
> >> &nbsp;
> >>
> >>[1] Reconstruction vs. Generation: Taming Optimization Dilemma in Latent Diffusion Models, CVPR 2025.

---

> > ### Comment · Reviewer_ciwU · 2025-08-05
> >
> > Thank the author for addressing issues raise in the previous comment. The pointed out figures are helpful for assessing the quality of the model when processing image with more various condition such as thicker subjects or side poses. Please include more in-the-wild cases in the revised manuscript as promised for better demonstrating the performance of the model. It would also be valuable to include a discussion of the identity-changing issue in the limitations section, as preserving identity is an important aspect of a virtual try-on framework, particularly for practical applications.

---

> > > ### Author Response · Authors · 2025-08-06
> > >
> > > Dear Reviwer,
> > >
> > > Thank you again for your constructive comments. As suggested,  we will include more qualitative results and discussions about identity-change issue in the revised revision if this paper is accepted.

---

### Official Review · Reviewer_5r1a · 2025-06-20

**Clarity:** 3
**Significance:** 2
**Originality:** 3
**Rating:** 4
**Confidence:** 3

**Summary:**

This paper presents a new training pipeline for virtual try-on (VTON) tasks. The pipeline involves multiple complicated stages that trying to find the most appropriate supervision signals for training a generative model while addressing Inherent-Occlusion and Acquired-Occlusion. In experiments, the proposed method was evaluated on 3 datasets against a total of 21 baselines.

**Questions:**

1. It seems that the mask-free approach serves as an additional image refinement step built upon the inpainted results from mask-based methods. Similarly, the proposed approach appears to be an extra refinement step based on a pre-trained mask-free model. Is this understanding correct? If so, direct comparisons with mask-based methods may not be strictly fair, given that those methods involve little to no refinement.
2. Why did the authors use SD1.5 instead of SDXL as the base generative model? BooW-VTON has shown that SDXL produces better results, so it is unclear why SDXL was not used.
3. In Section 4.2, Equation 10, the gating parameter $\gamma$ is determined solely by the sizes of the neck and arms. What is the rationale behind using only these two body parts?
4. The authors claim that their method generalizes to in-the-wild scenes. However, according to Section 5.2, the model was trained on the VITON-HD dataset and tested on the DressCode dataset, which is not strictly in-the-wild evaluation. I suggest including additional experiments on real-world images of humans and clothing sourced from the internet.

**Ethical Concerns:**

["NO or VERY MINOR ethics concerns only"]

**Final Justification:**

Thanks authors for the rebuttal. Some of my concerns have been addressed. While I am still concerned about the novelty&complexity of the method and the moderate visual results demonstrated, I think the paper with modifications based on rebuttals is ready to be present at NeurIPS.

**Limitations:**

Yes.

**Paper Formatting Concerns:**

No formatting concerns.

**Quality:**

3

**Strengths And Weaknesses:**

Strengths:
1. The presentation of the paper is pretty good. The problems are clearly defined, and the figures and tables are mostly clear and straightforward.
2. The experiments are comprehensive and solid. The model outperforms other methods across multiple benchmarks.
3. The appendix is thorough, with Algorithm 1 being particularly clear and helpful.

Weaknesses:
1. The technical contribution is incremental. While the proposed approach for identifying a more reliable training objective is valuable, it appears to be more of an engineering effort, as the optimization target remains training or fine-tuning a generative model. Moreover, the problems of Inherent Occlusion and Acquired Occlusion have already been defined in [16], which further diminishes the novelty of this work.
2. Although cloth/body occlusions are greatly improved based on the figures and supplementary materials, the generation quality of human faces and limbs is significantly worse than that of the baseline methods. The generated faces are often distorted and blurred. Is there a specific explanation for this?

See "questions" below for further doubts and minor weakness.

---

> ### Author Rebuttal · Authors · 2025-07-31
>
> We appreciate all your valuable suggestions and constructive comments.
>
> >**W1**: The technical contribution is incremental. While the proposed approach for identifying a more reliable training objective is valuable, it appears to be more of an engineering effort, as the optimization target remains training or fine-tuning a generative model. Moreover, the problems of Inherent Occlusion and Acquired Occlusion have already been defined in [16], which further diminishes the novelty of this work.
> >>**RW1**: We cannot agree with your comment that “The technical contribution is incremental” based on the reasoning that “training or fine-tuning a generative model appears to be more of an engineering effort”. As we know, “pre-training and fine-tuning” has already been a widely used paradigm to generalize the powerful foundation model that per-trained on large-scale datasets to the downstream tasks, such as virtual try-on. Since most downstream tasks have their own characteristics, designing a novel and effective fine-tuning strategy to address the problems arising in the generalization process is definitely a strong technical contribution. Moreover, although the problems of Inherent Occlusion and Acquired Occlusion have already been defined in [16], our work provides a more insightful analysis on the causes of these two types of occlusions (see Section 3). And after that, we designed a targeted solution based on the analysis, which is simple-yet-effective for addressing the occlusion problems in (mask-free) virtual try-on. Therefore, we believe that our proposed method has sufficient novelty.
>
> >**W2**: Although cloth/body occlusions are greatly improved based on the figures and supplementary materials, the generation quality of human faces and limbs is significantly worse than that of the baseline methods. The generated faces are often distorted and blurred. Is there a specific explanation for this?
> >>**RW2**: This is mainly due to the limitations of latent diffusion models. Encoding and decoding human images with a pre-trained KL-regularized autoencoder (e.g., VQGAN) in latent diffusion models will inevitably discard a certain amount of face details (especially around the eyes) [1]. Most existing virtual try-on methods often mitigate such information loss by performing the following post-processing step, which adopts an inpainting mask to paste the human identity area from the original image onto the same region of the generated result.
> >>
> >>$$p_{result} = m_{agn} \odot p + (1 - m_{agn}) \odot p_{un}$$
> >>However, since the inpainting mask used above may also be imprecise and introduce some new occlusions, we abandon this post-processing step in our proposed framework. Prompted by your comment, we will incorporate optimization of the latent diffusion model generation quality [1] in our future work to achieve further improvements.
> >>
> >> &nbsp;
> >>
> >>[1] Reconstruction vs. Generation: Taming Optimization Dilemma in Latent Diffusion Models, CVPR 2025.
>
> >**Q1**: It seems that the mask-free approach serves as an additional image refinement step built upon the inpainted results from mask-based methods. Similarly, the proposed approach appears to be an extra refinement step based on a pre-trained mask-free model. Is this understanding correct? If so, direct comparisons with mask-based methods may not be strictly fair, given that those methods involve little to no refinement.
> >>**RQ1**:  This is a misunderstanding. The mask-free and mask-based strategies are two different paradigms to address the unpaired data problem, and the greatest difference between them is that the mask-free methods **DO NOT** require to generate inpainting masks in the inference phase. To eliminating such requirement, the mask-free methods can synthesize pseudo reference images from a **pre-trained generative model (SD1.5 used in our implementation) or a mask-based method**.  So the mask-free approach is not an additional image refinement step for mask-based methods. In addition, our proposed method is built by **fine-tuning the SD1.5 diffusion model from scratch not on another pre-trained mask-free model**, so it does not serve as an extra refinement step. As shown in Figure 2, similar to other mask-free methods, our approach first synthesizes and selects the pseudo reference images and their corresponding occlusion-augmented person images. During this process, our designed background pre-replacement and covering-and-eliminating operations are employed to deal with inherent and acquired occlusions, respectively. After that, we fine-tune the SD1.5 diffusion model, which is a widely-used foundation model for virtual try-on tasks, by using the selected paired samples. According to the above explanation, since both mask-free and mask-based methods start from the same pre-trained diffusion model, and mask-free methods (including our approach) even do not utilize the inpainting mask information during the inference phase, we believe that the comparisons between our method and mask-based methods are sufficiently fair.
>
> >**Q2**: Why did the authors use SD1.5 instead of SDXL as the base generative model? BooW-VTON has shown that SDXL produces better results, so it is unclear why SDXL was not used.
> >>**RQ2**: To make fair comparison with existing virtual try-on benchmarks, we adopt their commonly used SD1.5 backbone. And moreover, due to the limited computational resources of our laboratory, it is infeasible to conduct experiments using SDXL.  Based on the above reasons, we use SD1.5 instead of SDXL as our base generative model. Nevertheless, the qualitative results in Figures 3 and 4 and the quantitative results in Table 2 demonstrate that, even with the relatively weaker SD1.5 backbone, our method outperforms SDXL-based counterparts such as BooW-VTON and IDM-VTON on several metrics. This provides sufficient evidence for the effectiveness of our approach.
>
> >**Q3**: In Section 4.2, Equation 10, the gating parameter $\gamma$ is determined solely by the sizes of the neck and arms. What is the rationale behind using only these two body parts?
> >>**RQ3**: We have explained the reason for using these two body parts in Lines 198–202 of our paper. The gating parameter $\gamma$ is used to indicate whether the garment $g_{un}$ fully covers $g$ or not. For the upper-body clothing, the body areas that are most likely to expose skin are the arms and neck. Therefore, we use these two body parts to generate $\gamma$ so as to determine the coverage of different garments.
>
> >**Q4**: The authors claim that their method generalizes to in-the-wild scenes. However, according to Section 5.2, the model was trained on the VITON-HD dataset and tested on the DressCode dataset, which is not strictly in-the-wild evaluation. I suggest including additional experiments on real-world images of humans and clothing sourced from the internet.
> >>**RQ4**: Considering the background augmentations used in our background pre-replacement operation, the proposed method should technically be able to generalize to in-the-wild images. In the first and last two rows of Figure 10 and the last two rows in Figure 11, we present some qualitative results with more practical backgrounds rather than plain colors. And moreover, the right block of Figure 8 also displays some results of heavier persons with sideways poses. Nevertheless, as suggested, we will add more results under real-world in-the-wild scenarios in our revised version to further validate the generalization ability of our method.

---

### Official Review · Reviewer_1L7U · 2025-07-01

**Clarity:** 2
**Significance:** 3
**Originality:** 3
**Rating:** 4
**Confidence:** 2

**Summary:**

This paper introduces a mask-free VTON framework, which includes background pre-replacement operation and covering-and-eliminating operation, to address the inherent and acquired occlusion problems in existing VTON methods. The background pre-replacement operation mitigates inherent occlusions by preventing the model from confusing target clothing information with the human body or image background. The covering-and-eliminating operation enhances the model’s ability to understand and model human semantic structures, thereby reducing acquired occlusions.

**Questions:**

1. From the visualizations in Fig.3 and in supplementary materials, I noticed that there exists common distortion in human models' face area, especifically around the eyes. In addition, the generated image resolution of the proposed method seems to be lower than comparison baselines. Do authors notice these issues, and what do you think might be the cause?

2. From the illustration of Background Pre-Replacement process in Figure 13, I notice that there still exists inherent occlusion issue, where the mask for right arm is not accurate. This leads to another question regarding the background pre-replacement process. It essentially depends on the quality of the segmentation mask, so how can it eliminate the inherent occlusion issue brought out by inaccurate masking?

**Ethical Concerns:**

["NO or VERY MINOR ethics concerns only"]

**Final Justification:**

Thanks the authors for the response. Some of my concerns have been addressed. Although I'm not so familiar with VTON area, this work provides an interesting and effective framework to resolve two critical occlusions, and provides promising results. I'll tend to keep my scores.

**Limitations:**

Please refer to previous sections.

**Quality:**

3

**Strengths And Weaknesses:**

Strengths of this paper include:
1. The motivation of the proposed method is pretty reasonable, as it deals with the major concerns of current VTON frameworks, which are inaccurate masking that missegments the human body, background, and clothing regions, as well as inaccurate representations that leads to erroneous human structure parsing.
2. The writing of this paper is well organized. It first introduces the two types of occlusions in detail, then proposes the method that addresses these two types of occlusions respectively.
3. Extensive quantitaive and qualitative experiments validate the effectiveness and generalization ability of the proposed method.

Please refer to next section for questions.

---

> ### Author Rebuttal · Authors · 2025-07-31
>
> We appreciate all your valuable suggestions and constructive comments.
>
> > **Q1:**  From the visualizations in Fig.3 and in supplementary materials, I noticed that there exists common distortion in human models' face area, especifically around the eyes. In addition, the generated image resolution of the proposed method seems to be lower than comparison baselines. Do authors notice these issues, and what do you think might be the cause?
> >> **R1:** Yes, we have noticed these issues. We discuss the causes of them as follows:
> >>
> >> **(1)** The reason of human face distortion is because that encoding and decoding human images with a pre-trained KL-regularized autoencoder (e.g., VQGAN) in latent diffusion models will inevitably discard a certain amount of face details (especially around the eyes) [1]. Most existing virtual try-on methods often mitigate such information loss by performing the following post-processing step, which adopts an inpainting mask to paste the human identity area from the original image onto the same region of the generated result.
> >>$$p_{result} = m_{agn} \odot p + (1 - m_{agn}) \odot p_{un}$$
> >>However, since the inpainting mask used above may also be imprecise and introduce some new occlusions, we abandon this post-processing step in our proposed framework. Prompted by your comment, we will incorporate optimization of the latent diffusion model generation quality [1] in our future work to achieve further improvements.
> >>
> >> **(2)** The lower image resolution is mainly caused by the fewer inference steps we adopt. In order to speed up the visualization process of the qualitative results of our method, we only run 20 denoising steps during inference while other comparison methods employ 50 denoising steps. If more inference steps are used, our method can generate better results with higher resolution.
> >>
> >> &nbsp;
> >>
> >>[1] Reconstruction vs. Generation: Taming Optimization Dilemma in Latent Diffusion Models, CVPR 2025.
>
> > **Q2:**  From the illustration of Background Pre-Replacement process in Figure 13, I notice that there still exists inherent occlusion issue, where the mask for right arm is not accurate. This leads to another question regarding the background pre-replacement process. It essentially depends on the quality of the segmentation mask, so how can it eliminate the inherent occlusion issue brought out by inaccurate masking?
> >> **R2:** This may be a misunderstanding. Our background pre-replacement operation is designed to address the problem of inaccurate segmentation masks. Take Figure 13 as an example, if without the background pre-replacement operation, the clothing in the unmasked areas of the right arm will be incorrectly preserved in the pseudo reference image. In the correct situation, such clothing information should be generated by the model, but now they can be simply maintained like the human body or background because they appear in both pseudo reference image and ground-truth image. Therefore, training with these samples can establish incorrect associations between the target clothing information and the human body or background pixels. In contrast, by using our background pre-replacement operation, we replace the inaccurate unmasked clothing areas in both pseudo reference image and ground-truth image with the same pure background or random scene image. By doing so, those areas will only contain the background information that should be maintained during the generation process, instead of target clothing pixels (as illustrated in Figure 2). Therefore, the clothing information in other accurately masked areas that need to be generated can be more effectively differentiated from the background, thus leading to better generation results.

---

> > ### Comment · Reviewer_1L7U · 2025-08-05
> >
> > Thank the author for their responses to my questions. I'm still a little bit confused about the issue that occurred in Fig.13. Please correct me if I'm wrong, but after the background pre-replacement operation, a small portion of the right forearm is still inaccurately unmasked and replaced by pure background. In such case, this area will be preserved but not generated by the network? Will it cause problems?

---

> > > ### Author Response · Authors · 2025-08-06
> > >
> > > Dear Reviewer,
> > >
> > > Thank you for your reply. Let me explain this issue step by step. If without our background pre-replacement operation, the unmasked right-arm cloth will appear in both ground-truth image and reference image. In this case, the model will only preserve this information instead of generating it. In other words, the model treats the unmasked right-arm cloth as the background or human body. However, the cloth in this area has a similar appearance with those in other cloth-wearing areas. This will result in that the model cannot distinguish the garment information that should be generated from the **real** background information that should be preserved, and establish incorrect associations between them. Based on this analysis, our background pre-replacement operation attempts to mitigate these spurious correlations by replacing the unmasked right-arm cloth with a pure background or a random scene image. In this way, such unmasked area is filled with background pixels instead of clothing pixels. You can think of this process as a situation that the human body in the original image is partially obscured by an object (often seen in in-the-wild images), thus this object should be maintained during the generation and will not introduce nagtive effects. This will explicitly separate the garment to be generated from the replaced background, and promote the model to capture correct garment information, thus improving its generative performance. More importantly, our background pre-replacement operation is only employed during the training phase to **synthesize more effective pseudo reference images**, so it will not affect the inference stage.

---

> > > ### Author Response · Authors · 2025-08-09
> > >
> > > Dear Reviewer,
> > >
> > > We sincerely appreciate your professional review. We have responded to your additional questions. If possible, would you please check our response? We hope it has addressed your concerns. If any questions remain, please feel free to let us know. Thank you for your time.
> > >
> > > Authors of Submission 5367

---

### Official Review · Reviewer_yELi · 2025-07-03

**Clarity:** 3
**Significance:** 3
**Originality:** 3
**Rating:** 5
**Confidence:** 3

**Summary:**

This is a virtual try-on paper focusing on scenarios where occlusion appears in images. More specifically, the authors group the occlusion-related problems into two main categories: (1) inherent, and (2) acquired. Main contributions are: (1) the analysis and formulation of the occlusion-related problems, (2) novel pre-replacement operation to prevent inherent occlusions, (3) novel covering-and-eliminating procedure to prevent acquired occlusions, (4) comparisons against many other methods (useful as a benchmark).

**Questions:**

I think that showing and discussing limitations even further in this work would be useful for the community. The paper already identifies and suggests potential solutions to two interesting limitations. Can the authors include more qualitative results that depict the failures of the proposed method (showing a larger variety of clothing, body shapes, and poses)?

It seems that in Figure 5 (GANs), there are some artifacts in the results. Can the authors explain why? Is there a significant difference in the results for GANs as compared to diffusion models?

Can the authors introduce some qualitative results with more complex backgrounds?

**Ethical Concerns:**

["NO or VERY MINOR ethics concerns only"]

**Final Justification:**

After reading the other reviews and the rebuttal, I decided to keep my Accept score. My score could have been higher if the paper had provided a more systematic overview of the limitations of existing methods, including its own (and comparisons on such extreme cases). I believe papers that clearly expose such limitations can have a greater impact, and this work moves in that direction, which I found both interesting and useful.

**Limitations:**

yes

**Paper Formatting Concerns:**

/

**Quality:**

3

**Strengths And Weaknesses:**

Strengths:

The paper is intuitive and provides insights into where the current methods are failing and provides potential solutions to those problems.

The method produces most detailed results (according to the perceptual error (LPIPS)) in Table 2 and maintains high ranking on other metrics in comparison with many other methods (useful as a benchmark).


Weaknesses:

Quality of the results: In Figure 5 there are still seams/mismatches in the “GANs + Ours” column.

Data: only standard fashion datasets were used, limiting the variety of clothing, body shapes, and poses.

There are no examples of results with more complicated backgrounds even though the model technically should be able to handle such examples due to the background augmentations that are used. Also see Line 305: “The results in Fig. 6 show that our method can better handle person images with diverse background styles” -> Fig. 6 contains examples with mostly white backgrounds only.

The paper does not contain many examples of failure.

There are some typos, for example: Line 165 aim -> aims.

---

> ### Author Rebuttal · Authors · 2025-07-31
>
> We appreciate all your valuable suggestions and constructive comments.
>
> > **W1**: Quality of the results: In Figure 5 there are still seams/mismatches in the "GANs + Ours" column.  **And Q2**: It seems that in Figure 5 (GANs), there are some artifacts in the results. Can the authors explain why? Is there a significant difference in the results for GANs as compared to diffusion models?
> >> **R1**: As shown in Figure 5,  there are some artifacts observed in the experimental results of both "GANs" and "GANs+Ours". But when we apply our method to a diffusion-based backbone (SD 1.5 used in this work), we can find that these artifacts are eliminated (see our results in Figure 3 and 4). So we believe that such problem lies in the relatively weaker generative capabilities of the GANs backbone compared to diffusion models. The adversarial training process of GANs is usually unstable and may introduce those undesirable artifacts into the final results. In addition, the quantitative comparisons in Table 2 and 3 also demontrate that the diffusion-based methods consistently achieve the best and second-best results across all evaluation metrics, outperforming the GANs-based benchmarks. Therefore, there is a significant difference in results between diffusion models and GANs.
>
> > **W2**: Data: only standard fashion datasets were used, limiting the variety of clothing, body shapes, and poses.
> >> **R2**: To ensure a fair comparison with existing methods, we employ the same datasets as them for our performance evaluation. The DressCode dataset, in particular, comprises 53,792 garment images spanning three categories—upper body, lower body, and dresses—and an equal number of human images featuring diverse skin tones, body shapes, and poses. Therefore, achieving state-of-the-art results in both quantitative and qualitative experiments on this dataset, especially in handling occlusions, is strong evidence of our method’s effectiveness and its advantages over existing approaches.
>
> > **W3**: There are no examples of results with more complicated backgrounds even though the model technically should be able to handle such examples due to the background augmentations that are used. Also see Line 305: “The results in Fig. 6 show that our method can better handle person images with diverse background styles” -> Fig. 6 contains examples with mostly white backgrounds only. **And Q3**: Can the authors introduce some qualitative results with more complex backgrounds?
> >> **R3**: In the first and last two rows of Figure 10 and the last two rows in Figure 11, we present some qualitative results with more complex backgrounds rather than plain colors. We can see that our method can handle these background styles well. Moreover as suggested, in the revised version of this paper, we will show more results on these backgrounds and introduce other results under in-the-wild scenario in Figure 6 to support our claim.
>
> > **W4**: The paper does not contain many examples of failure. **And Q1**: I think that showing and discussing limitations even further in this work would be useful for the community. The paper already identifies and suggests potential solutions to two interesting limitations. Can the authors include more qualitative results that depict the failures of the proposed method (showing a larger variety of clothing, body shapes, and poses)?
> >> **R4**:  We have already discussed the limitations and future work in Section F of our Appendix and have displayed a failure case in Figure 12. Nevertheless, as suggested, we will add more examples of failure in our revised version to better illustrate the current limitations of the proposed method and guide our future work.
>
> > **W5**: There are some typos, for example: Line 165 aim -> aims?
> >> **R5**:  Thanks for pointing out this. We have carefully checked our paper and will correct all such typos (including L165: aim -> aims) in the revised version.

---

> > ### Comment · Reviewer_yELi · 2025-08-05
> >
> > Thank you for the rebuttal. I understand that the new guidelines restrict adding more qualitative examples during the rebuttal phase. However, I would still recommend including a systematic overview of the method's limitations, failure cases, and extreme scenarios in the main paper. I will keep my score unchanged.

---

> > > ### Author Response · Authors · 2025-08-06
> > >
> > > Dear Reviwer,
> > >
> > > Thank you again for your valueable suggestions and constructive comments. As suggested, if this paper is accepted, we will include more qualitative results of failure cases and those obtained in extreme scenarios, as well as some discussions about potential limitations in the revised revision.

---

### Decision · Program_Chairs · 2025-09-17

**Decision:**

Accept (poster)

**Comment:**

This paper presents a valuable contribution to the virtual try-on field by explicitly addressing the underexplored but critical challenge of occlusions. The work is commendable for its systematic approach to the problem, clearly categorizing occlusion types into "inherent" and "acquired," which provides a useful conceptual framework for future research.

The proposed technical solutions are well-motivated by the identified limitations of current methods. The novel pre-replacement operation for inherent occlusions and the covering-and-eliminating procedure for acquired occlusions represent a logical and intuitive approach to mitigating these distinct issues. The experimental results are persuasive, demonstrating that the method achieves state-of-the-art performance on key perceptual metrics like LPIPS while maintaining strong performance on other standard benchmarks. The extensive comparisons against numerous other methods add significant value to the paper as a useful benchmark for the community.

While the paper has weaknesses—such as residual artifacts in some results, the use of standard datasets limiting variety, and a lack of highly complex background examples—these do not outweigh its core contributions. The identified limitations are common in the field and are often addressed in subsequent work. The paper's foundation is solid, and its novel focus on occlusion provides a clear pathway for advancement. The issues mentioned, such as dataset scope and minor typos, can be readily addressed in a final camera-ready version.

In conclusion, this paper introduces a timely and insightful approach to a key problem in virtual try-on. The proposed method is effective, the experiments are thorough, and the conceptual framing of occlusions is itself a contribution. The strengths significantly outweigh the weaknesses, making this paper a strong candidate for acceptance.